# Electrostatic anchoring precedes stable membrane attachment of SNAP25/SNAP23 to the plasma membrane

Pascal Weber[1], Helena Batoulis[1], Kerstin M Rink[2], Stefan Dahlhoff[1], Kerstin Pinkwart[1], Thomas H Söllner[2], Thorsten Lang[1]*

[1]Membrane Biochemistry, Life and Medical Sciences (LIMES) Institute, University of Bonn, Bonn, Germany; [2]Heidelberg University Biochemistry Center (BZH), Heidelberg, Germany

**Abstract** The SNAREs SNAP25 and SNAP23 are proteins that are initially cytosolic after translation, but then become stably attached to the cell membrane through palmitoylation of cysteine residues. For palmitoylation to occur, membrane association is a prerequisite, but it is unclear which motif may increase the affinities of the proteins for the target membrane. In experiments with rat neuroendocrine cells, we find that a few basic amino acids in the cysteine-rich region of SNAP25 and SNAP23 are essential for plasma membrane targeting. Reconstitution of membrane-protein binding in a liposome assay shows that the mechanism involves protein electrostatics between basic amino acid residues and acidic lipids such as phosphoinositides that play a primary role in these interactions. Hence, we identify an electrostatic anchoring mechanism underlying initial plasma membrane contact by SNARE proteins, which subsequently become palmitoylated at the plasma membrane.

*For correspondence: thorsten. lang@uni-bonn.de

**Competing interests:** The authors declare that no competing interests exist.

## Introduction

Palmitoylation is a post-translational modification of a protein which causes its stable attachment to a cellular membrane. Examples of proteins that follow this paradigm are the homologous SNARE (soluble N-ethylmaleimide-sensitive factor attachment receptor) proteins SNAP25 and SNAP23, which after translation are initially cytosolic proteins. In order to function in vesicle fusion, they relocate to the plasma membrane. SNAP23 is ubiquitously expressed, whereas the neuronal SNAP25 is highly abundant in the synapse and in the plasma membrane of neuroendocrine cells (*Jahn and Fasshauer, 2012*; *Wilhelm et al., 2014*; *Knowles et al., 2010*). Stable attachment to membranes is achieved after palmitoylation of a cysteine cluster, which is most probably catalyzed by the plasma membrane resident palmitoyl acyltransferase DHHC2. DHHC2 is characterized by the presence of a conserved DH(H/Y)C motif and can palmitoylate SNAP25 and SNAP23 (*Greaves et al., 2010*).

The majority of SNAP25 molecules reside in the plasma membrane, while 20% are located in a perinuclear recycling endosome-trans-Golgi network (*Aikawa et al., 2006*). A two-compartment model for SNAP25 trafficking has been proposed which speculates that the endocytic recycling of SNAP25 might be coupled to its depalmitoylation, followed by its repalmitoylation and recycling back to the plasma membrane (*Aikawa et al., 2006*). In any case, in steady-state, the large majority of SNAP25 molecules are stably attached to the cell membrane.

The minimal domain necessary for SNAP25 plasma membrane targeting has been mapped to amino acids 85–120 (*Gonzalo et al., 1999*), comprising the cysteine cluster at the N-terminus (for relevant SNAP25 region see *Figure 1*), while the C-terminus interacts with DHHC proteins

**eLife digest** Cells often communicate with each other by releasing chemicals that normally are stored in small membrane-bound compartments called vesicles. For example, when a neuron is stimulated, vesicles merge with its cell membrane and release their content into a gap between itself and other neurons. This complicated process involves many steps and molecules, including proteins called SNAREs.

Some SNARE proteins reside at the inner side of the cell membrane and help vesicles to fuse with this membrane. Two SNARE proteins called SNAP25 and SNAP23 are produced in the liquid inside the cell and initially float freely. Eventually, these proteins become directly anchored to the cell membrane, however, not much is known about what happens to these proteins in between these stages, or how they first attach to the membrane before anchoring to it.

Electrostatic forces between oppositely charged molecules are known to be important for them to bind with each other. Here, electrostatic forces are less likely to occur because SNAP25 and SNAP23 are both mostly negatively charged, and should therefore be repelled from the cell membrane, which also typically has a negative charge. However, both SNAP25 and SNAP23 have a small cluster of positively charged amino acids (the building blocks of proteins) near the attachment site, and Weber et al. have now tested whether this charge is sufficient to overcome the predicted repulsion.

The approach involved making mutant proteins with either more or less positively charged attachment regions. Mutant SNAP25 or SNAP23 proteins with more positive charges may stick more tightly but not necessarily more permanently to the membrane. However, when the number of positive charges was lowered, more of the proteins remained floating freely in the liquid inside the cell. These results suggest that even a small number of positively charged amino acids is sufficient to help a protein bind to a cell membrane for further processing.

The findings of Weber et al. reveal an early step in the life cycle of SNAP25 and SNAP23 before they anchor to the cell membrane. They suggest that finely tuned protein electrostatics can regulate how long a protein spends at a specific site and thereby indirectly determine its fate. Such fine-tuned protein electrostatics are difficult to recognize and could represent an underestimated regulatory mechanism in all types of cells.

(*Gonzalo et al., 1999*; *Greaves et al., 2010*). For DHHC interactions to occur, proximity to the membrane is an important factor and could even be rate limiting in the attachment process.

SNAP25 and SNAP23 are not modified in the cytosol by isoprenyl- or myristoyl groups, which would increase their membrane affinity and facilitate initial membrane contact; neither receptors nor membrane-targeting motifs have been identified previously and thus there has been some debate over how this initial contact may be mediated.

Polybasic amino acid patches are known to mediate non-specific interactions with anionic lipids. For instance, plasma membrane targeting of myristoylated K-Ras requires an N-terminal polybasic domain (*Cadwallader et al., 1994*; *Wright and Philips, 2006*), which localizes G protein α subunits to the plasma membrane, although this region is not required for subunit palmitoylation (*Pedone and Hepler, 2007*; *Crouthamel et al., 2008*). Other examples suggest that phosphorylation of basic residues located upstream of palmitoylated cysteines regulates the palmitoylation of a potassium channel through an electrostatic switch (*Jeffries et al., 2012*). In some instances, polybasic patches are the main driving force for protein attachment to the negatively charged plasma membrane (*Cho and Stahelin, 2005*).

SNAP25 has a net negative charge: of its 206 amino acids, 21% are negatively and 14% are positively charged. Still, a modest excess of three positive charges around the cysteine cluster might be available for non-specific interactions with anionic lipids. Nevertheless, this charge accumulation appears very small when compared to the number of positive charges that mediate electrostatic contacts in other cellular processes (*Heo et al., 2006*). Here, we set out to investigate whether the subcellular distribution of SNAP25 and SNAP23 can be regulated through such a small accumulation of positive charges, despite the proteins' overall negative net charge.

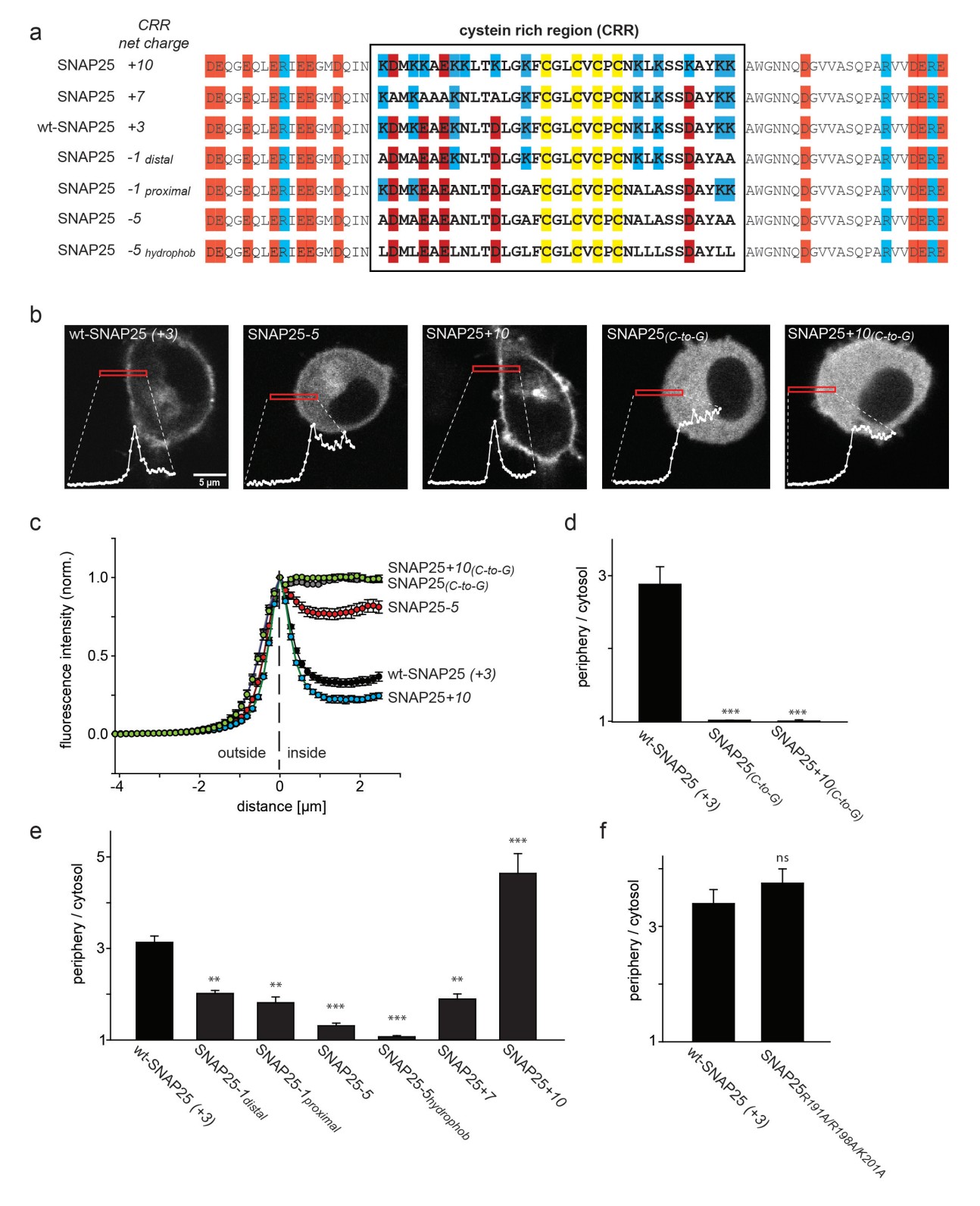

**Figure 1.** SNAP25 plasma membrane targeting in live cells. (**a**) Diagram of the amino acid sequences from position 51 to 125 for wild type SNAP25 and for several constructs in which the net charge of the cysteine rich region (CRR; box) is increased or decreased. Numbers associated with the constructs' names refer to the net charge of the CRR. Cysteines are highlighted in yellow; negatively and positively charged amino acids are highlighted in red and blue, respectively. A net charge of +3 is found in all mammalian SNAP25 proteins whose encoding genes are available in the UniProtKB/*Swiss*-Prot-data

*Figure 1 continued on next page*

*Figure 1 continued*

bank (see *Figure 1—figure supplement 1*). (**b**) Confocal micrographs from live PC12 cells expressing wt-SNAP25 (+3), SNAP25–*5* and SNAP25+*10* as N-terminally GFP-tagged constructs. Also shown are two more constructs, based on wt-SNAP25 (+3) and SNAP25+*10*, in which the four cysteines for palmitoylation are exchanged for glycines (SNAP25$_{(C-to-G)}$ and SNAP25+*10*$_{(C-to-G)}$). Red elongated boxes mark the regions of interest (ROIs) in which the fluorescence distribution at the cell periphery was analysed by linescans. White graphs illustrate the corresponding fluorescence traces. (**c**) For one experiment several traces were averaged. (**d–f**) Ratio between cell periphery and cytosol signal for (**d**) the variants lacking cysteines for palmitoylation, (**e**) constructs with an altered charge around the cysteine cluster, and (**f**) a construct with an eliminated polybasic cluster located at the C-terminus of SNAP25. Values are given as means ± S.E.M. (n = 3–19; t-test *p<0.05, **p<0.01, ***p<0.001, ns = not significant). For the constructs exhibiting weakest and strongest targeting in (**c**), *Figure 1—figure supplement 2* shows that the ratio between cell periphery and cytosol signal is independent of the expression level.

The following figure supplements are available for figure 1:

**Figure supplement 1.** Cysteine-rich regions from different species.

**Figure supplement 2.** No correlation between the periphery/cytosol signal ratio and the expression level.

# Results

## Analysis of SNAP25 targeting in live cells

We wondered whether the above mentioned small charge accumulation close to the cysteine cluster is exclusively found in rat SNAP25. Comparison of the cysteine-rich region of the SNAP25 protein (UniProtKB/*Swiss*-Prot-data bank) in a variety of species revealed that this region carries a net positive charge in all species except *Drosophila* (*Figure 1—figure supplement 1*). The distribution of charged amino acids adds up to +3 in all seven mammalian species, and also in chicken, zebrafish and goldfish (although for zebrafish and goldfish there is also an isoform with a charge of +1). The cysteine-rich region has four cysteines for palmitoylation in the centre, which are flanked by four lysines on each side (*Figure 1a*). The sequence around the cysteines also contains five acidic amino acids: four upstream and one downstream of the cysteines. Hence, eight basic and five acidic amino acids yield a net charge of +3 located downstream of the cysteine cluster.

If positive charges close to the cysteine cluster (*Figure 1a*) constitute the main driving force for initial plasma membrane association, their elimination should diminish plasma membrane targeting. To test this hypothesis, we overexpressed GFP-tagged SNAP25 in neuroendocrine PC12 cells and analysed how plasma membrane targeting depends on these charges. Equatorial optical sections were imaged in live cells (*Figure 1b*). Linescans perpendicular to the plasma membrane reveal the GFP-SNAP25 distribution at the cell periphery (plasma membrane + cytosol that optically cannot be resolved) and the cytosol (*Figure 1b and c*). Relating these values to each other yields a ratio >1 if there is a plasma membrane-associated fraction.

A variety of SNAP25 mutants with increased or decreased charge in the cysteine-rich region were analysed. As control for non-targetable protein, cysteine palmitoylation sites were substituted by glycines (*Figure 1d*). As expected, SNAP25 proteins that lacked cysteines (construct SNAP25$_{(C-to-G)}$) did not locate to the plasma membrane (*Figure 1d*). We then reduced the charge from wt-SNAP25 (+3) to −1 by substituting four lysines with alanines (for construct details, see *Figure 1a*). In SNAP25-*1*$_{distal}$, the outer four lysines up- and downstream of the cysteine cluster are exchanged, and in SNAP25-*1*$_{proximal}$ the four inner ones are exchanged. Both of these constructs showed strongly diminished plasma membrane targeting (*Figure 1e*). Further reducing positive charges to a net charge of −5, by substituting all eight lysines with alanines (SNAP25-*5*; see *Figure 1a*), abolishes targeting almost completely (*Figure 1e*).

We also tested a construct termed SNAP25-*5*$_{hydrophob}$ which is identical to SNAP25-*5* except that the lysines were not replaced by alanines but instead by more hydrophobic leucines (*Figure 1a*). The reasoning for increasing the hydrophobicity is a previous report suggesting that the hydrophobicity of this domain plays a role in initial membrane association (*Greaves et al., 2009*). Compared to SNAP25-*5*, SNAP25-*5*$_{hydrophob}$ showed no increase in targeting; instead, targeting was further reduced and hardly detectable.

We next tested whether more positive charges would increase targeting efficiency. Here, the finding was ambiguous, as increasing the charge to +7 (SNAP25+7) or +10 (SNAP25+10; for constructs see *Figure 1a*) diminishes or promotes targeting, respectively (*Figure 1e*). Substituting the cysteines by glycines in the SNAP25+10 mutant (SNAP25+10_(C-to-G)) causes a cytosolic distribution (*Figure 1d*). Hence, an increase in positive charges cannot substitute for attachment by palmitoylation.

There is another small cluster of positive charges downstream of the cysteine-rich region in the C-terminal part of the SNAP25 linker. Elimination of these charges by introducing the mutations R191A, R198A and K201A has no effect on membrane targeting (*Figure 1f*). This indicates that the mere presence of positive charges is not sufficient for membrane targeting. Rather, the position of charged residues within the protein structure determines their effect.

SNAP23 is 60% identical to SNAP25. Like SNAP25, it carries three positive charges in the cysteine-rich region (*Figure 2a*). Reduction of the positive charges by six units in SNAP23-3 diminishes membrane association (*Figure 2*). As observed for SNAP25, increase of positive charges can promote (see SNAP23+10 in *Figure 2* and SNAP23+16 in *Figure 2—figure supplement 1*) or diminish (see SNAP23+11a and SNAP23+11b in *Figure 2—figure supplement 1*) membrane targeting. Finally, exchange of cysteines for glycines (SNAP23_(C-to-G)) yields a cytosolic distribution (*Figure 2*), showing that palmitoylation is also required for stable attachment of SNAP23.

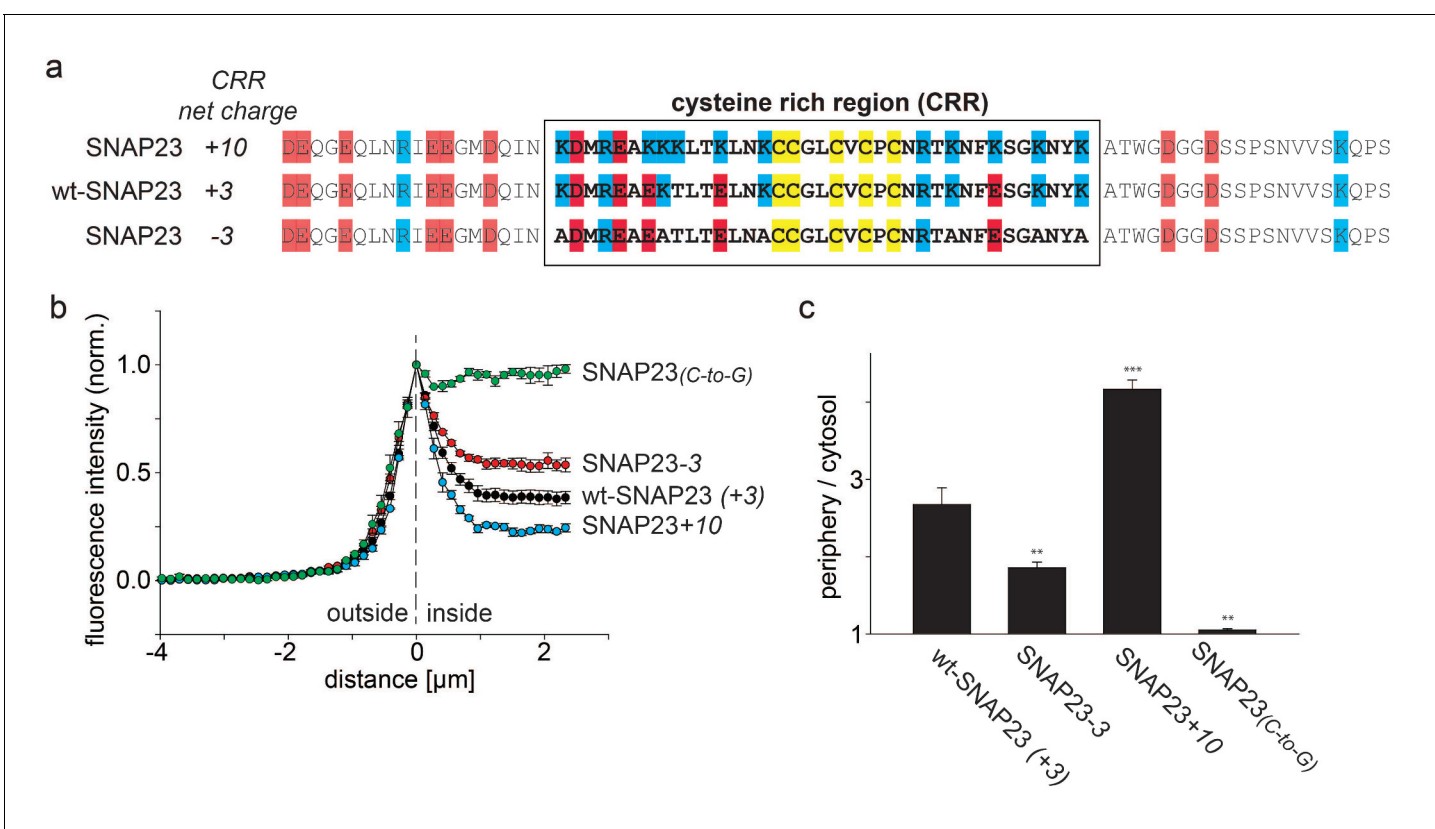

**Figure 2.** SNAP23 plasma membrane targeting. (a) Amino acid sequences from positions 45 to 119 shown for SNAP23+10, wt-SNAP23 (+3) and SNAP23-3. The box indicates the cysteine-rich region (CRR) in which mutations were introduced. Red, blue and yellow, respectively, highlight negatively charged amino acids, positively charged amino acids and cysteines. (b, c) The periphery/cytosol signal ratio from confocal micrographs was analysed as described in *Figure 1*. Values are given as means ± S.E.M. (n = 3–8; t-test *p<0.05, **p<0.01, ***p<0.001). For more SNAP23 constructs, see *Figure 2—figure supplement 1*.

The following figure supplement is available for figure 2:

**Figure supplement 1.** Correlation between SNAP23 targeting and charge of the cysteine-rich region.

In conclusion, the analysis of SNAP25/SNAP23 constructs shows that membrane targeting is dependent on positive charges located close to the cysteine cluster. Charges > +3 can increase or decrease targeting, suggesting that primary structure is not the only determinant for stronger targeting and that secondary structural elements are also important. Structural features may define whether the charges can be exposed to the membrane environment. Alternatively, they may determine the geometry of the electrostatic contact (and hence the accessibility of the cysteines) that plays a role in the palmitoylation reaction.

## SNAP25-association with isolated membranes

In addition to immunofluorescence imaging, we employed an independent method to analyse the subcellular distribution. SNAP25 constructs that showed strong effects in live cells and the constructs without palmitoylation sites were studied by cell fractionation (*Figure 3*). In contrast to the imaging analysis, the biochemical cell fractionation experiment makes it possible to relate the absolute amounts of the membrane and cytosol fraction to each other. In line with the imaging experiments, we observed that SNAP25-*5* has a weaker and SNAP25+*10* a stronger membrane association than wt-SNAP25 (*+3*). In the imaging experiment, no membrane-associated fraction of SNAP25(C-to-G) or SNAP25+*10*(C-to-G) was visible. This is different in the membrane fractionation assay where we detected a small membrane-associated fraction for SNAP25+*10*(C-to-G) (*Figure 3*). This fraction might have been overlooked by confocal microscopy, which is unable to resolve small plasma membrane-associated pools in the presence of a strong cytosolic background. For a better microscopic analysis, it was necessary to eliminate the cytosolic background. We therefore turned to 'unroofed cells'

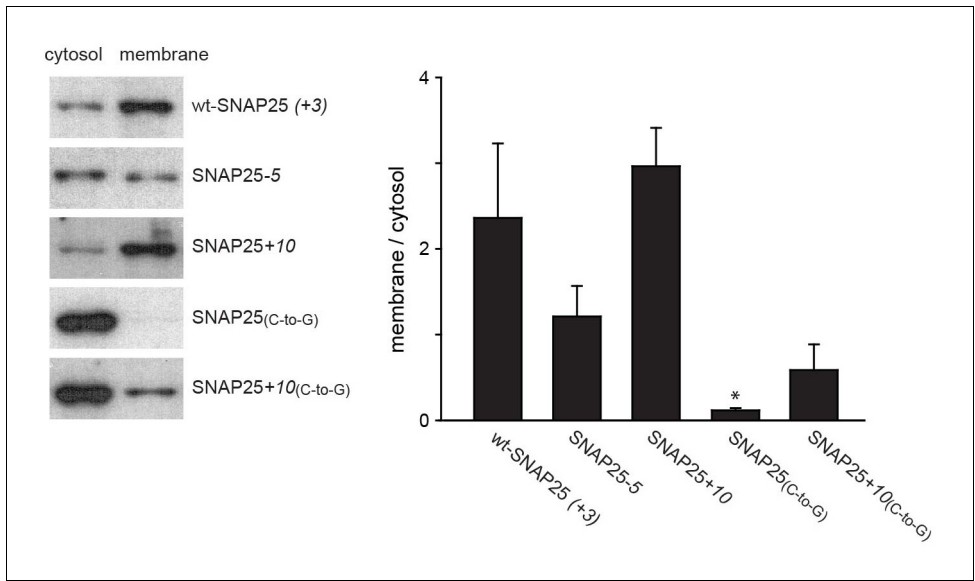

**Figure 3.** Subcellular distribution of SNAP25 constructs analysed by cell fractionation. PC12 cells expressing the indicated constructs were mechanically homogenised followed by centrifugation, yielding supernatant and pellet that contain the cytosolic and the membrane fraction, respectively. Fractions were analysed by Western blotting using an antibody against GFP. Left: immunoblots of one representative experiment. For each construct, the respective cytosol and membrane fractions are shown at arbitrary scaling (for the entire blot see *Figure 3—figure supplement 1*; for the average expression levels see *Figure 3—figure supplement 2*). Right: the ratio between membrane-associated and cytosolic protein was quantified from the band intensities. Values are given as means ± S.E.M. (n = 4; t-test *p<0.05, **p<0.01, ***p<0.001).

The following figure supplements are available for figure 3:

**Figure supplement 1.** Entire Western blot.

**Figure supplement 2.** Variation of expression levels.

(*Heuser, 2000*), also called plasma membrane sheets. To achieve this, cells were exposed to a brief ultrasound pulse that exerts a shearing force to the upper cellular parts, leaving behind the basal plasma membranes. Then, the membrane sheets were imaged to quantify the recruited protein in the absence of the cytosolic background. Wt-SNAP25 (+3), SNAP25-5 and SNAP25+10 were readily detected on the membrane sheets (*Figure 4*) in amounts that would be expected from the confocal microscopic subcellular distribution analysis. SNAP25$_{(C-to-G)}$ was not detectable, either because it is present at undetectable amounts after membrane sheet generation (see also the hardly detectable membrane fraction of SNAP25$_{(C-to-G)}$ in the cell fractionation assay in *Figure 3*), or because it binds weakly and washes off during sample mounting and imaging (which takes up to ≈ 35 min in total). By contrast, SNAP25 +10$_{(C-to-G)}$ was clearly present, despite the absence of palmitoylation sites, documenting a higher plasma membrane affinity when compared to SNAP25$_{(C-to-G)}$.

## Assessment of palmitoylation

Cell fractionation and membrane sheet experiments both show that an increase in positive charge produces a membrane-associated fraction even in the absence of palmitoylation. This poses the question of whether the small increase in membrane targeting of SNAP25+10 is, in part, achieved independently from stable membrane-attachment through palmitoylation.

To clarify this issue, we examined whether the extent of plasma membrane targeting correlates with the degree of palmitoylation for wt-SNAP25 (+3), SNAP25-5 and SNAP25+10, using SNAP25$_{(C-to-G)}$ as a negative control. Freshly transfected cells were incubated with a palmitate analogue carrying an alkyne-group to allow click-labelling with a fluorophore. Hence, the newly synthesized SNAP25 constructs should be palmitoylated by the clickable palmitate. Cells were lysed and the constructs were immunoprecipitated using their GFP-tags. A Cy5-fluorophore was covalently attached to the alkyne group through click-chemistry and the immunoprecipitate was subjected to western blot analysis. The membrane was immunostained for GFP, and the palmitate-Cy5 fluorescence was related to the GFP signal. As expected, the data show that SNAP25-5 is much less palmitoylated than wt-SNAP25 (+3), and SNAP25$_{(C-to-G)}$ shows no detectable traces of clickable-palmitates (*Figure 5*). SNAP25+10 is not significantly more palmitoylated than wt-SNAP25 (+3) (*Figure 5*).

The outcome of this experiment is well in line with (i) the linescan analysis (*Figure 1*), (ii) the biochemical cell fractionation assay (*Figure 3*) and (iii) the membrane sheet association assay (*Figure 4*). The four different test systems indicate that SNAP25-5 is less targeted to the plasma membrane and less palmitoylated than wt-SNAP25 (+3). Moreover, SNAP25$_{(C-to-G)}$ cannot associate with the membrane. Finally, SNAP25+10 shows a trend towards increased targeting. However, the data suggest that SNAP25+10 associates with the plasma membrane not only through palmitoylation but also by pure protein electrostatics.

## Effect of mutations on the interaction with syntaxin

Initial plasma membrane association via protein electrostatics may not be the only mechanism affected by the introduced mutations. A disturbed interaction with palmitoyl transferases is unlikely as all of the substitutions are located distal to the QPARV motif which is important for the interaction of SNAP25 and DHHC palmitoyl transferases (*Greaves et al., 2010*). However, some mutations map to the N-terminal SNARE motif of SNAP25 that interacts with plasmalemmal syntaxin 1A. Early studies on SNAP25 plasma membrane targeting proposed syntaxin as the receptor for initial contact establishment (*Vogel et al., 2000*; *Washbourne et al., 2001*), although a subsequent study provided compelling evidence against this hypothesis (*Loranger and Linder, 2002*).

To test whether our mutations affect the formation of a complex with syntaxin, we employed a fluorescence recovery after photobleaching (FRAP) assay. This assay is capable of probing interactions between SNAP25 and syntaxin by measuring the mobility of SNAP25 (*Halemani et al., 2010*). Mobility diminishes the stronger SNAP25 binds to syntaxin and the more syntaxin is present. The slow-down is dependent on a syntaxin–SNAP25 interaction, as it requires the N-terminal SNARE motif of SNAP25. Moreover, a mutant carrying introduced prolines in the N-terminal SNARE-motif of SNAP25 (prolines interfere with the alpha-helix formation, which in turn is required for SNARE-complex formation) moves almost independently of the syntaxin concentration (*Halemani et al., 2010*). We tested our constructs on membrane sheets with or without co-expressed syntaxin 1A-RFP. Syntaxin 1A-RFP slows down the mobility of wt-SNAP25 (+3), SNAP25-5 and SNAP25+10 in a

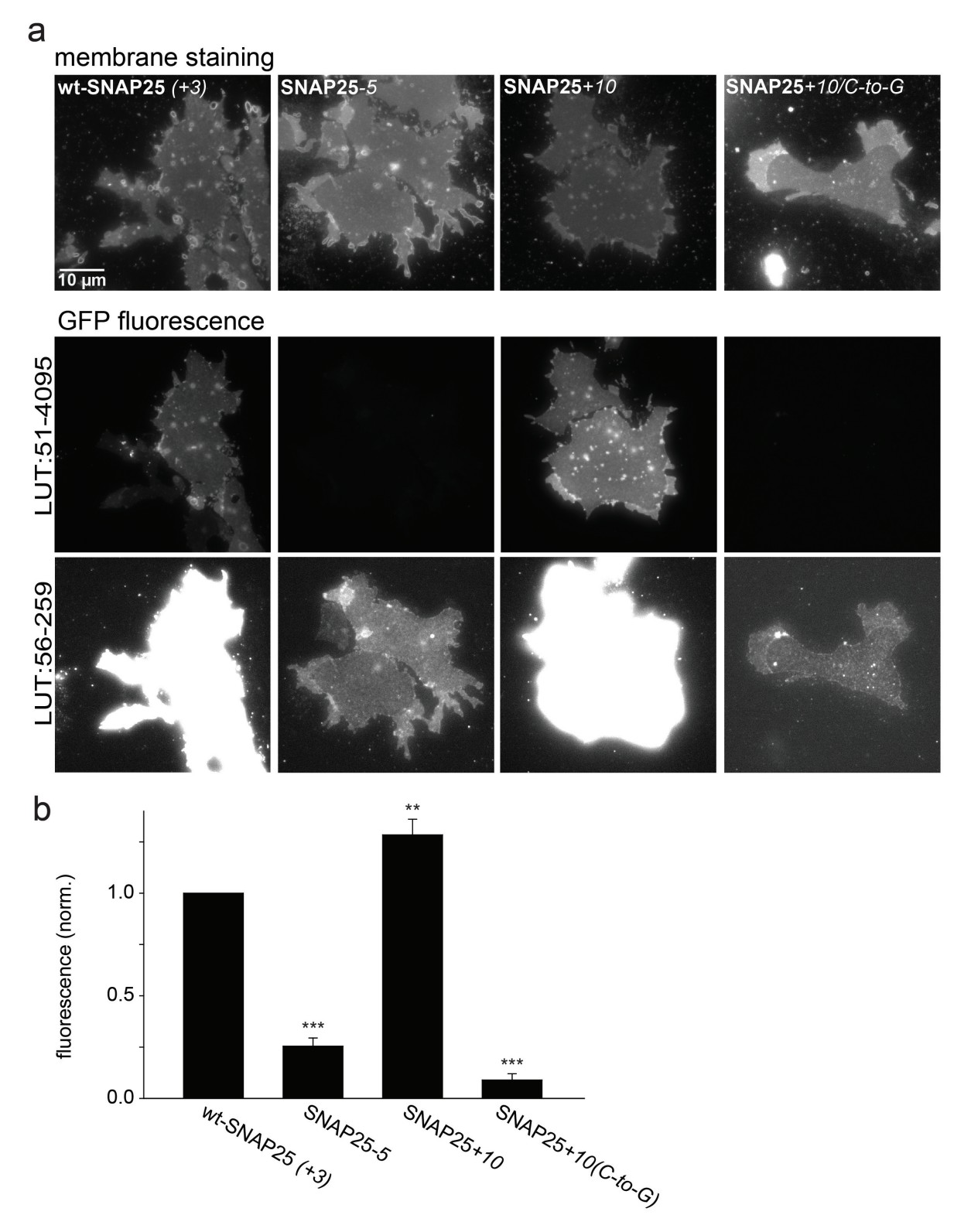

**Figure 4.** Association of wt-SNAP25 (+*3*), SNAP25-*5*, SNAP25+*10* and SNAP25+*10*$_{(C-to-G)}$ with isolated plasma membranes. (**a**) Plasma membrane sheets were generated from cells expressing the indicated GFP-SNAP25 constructs by mechanical shearing forces, followed by direct imaging. During imaging, the sample was screened for green fluorescence and all membranes exhibiting green fluorescence were imaged in the green channel, followed by imaging of the blue channel. We also analysed sheets from cells transfected with SNAP25$_{(C-to-G)}$, but in these samples, no green

*Figure 4 continued on next page*

*Figure 4 continued*

fluorescence was visually detectable in the screening process. Top, blue fluorescent dye (TMA-DPH) visualizing the location and shape of the membrane sheets; bottom, GFP fluorescence of the membrane sheet associated SNAP25 variants. The same images are shown at two different lookup tables (LUT). (b) Quantification of GFP-fluorescence on membrane sheets, normalized to wt-SNAP25 (+3). Values are given as means ± S.E.M. (n = 3–7; t-test *p<0.05, **p<0.01, ***p<0.001).

similar fashion, arguing against an altered capability of complex formation caused by the introduced mutations (*Figure 6*). Therefore, we conclude that the differences in initial membrane association are not due to an altered affinity of binding to syntaxin.

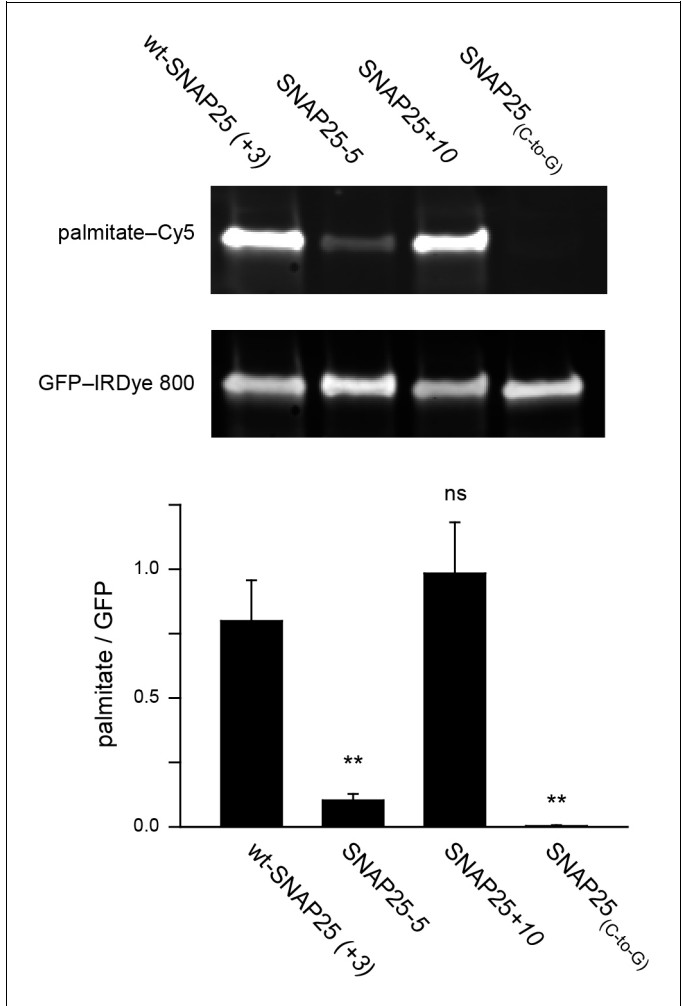

**Figure 5.** Assessment of the palmitoylation of GFP-SNAP25 constructs. PC12 cells were transfected with wt-SNAP25 (+3), SNAP25-5, SNAP25+10, or SNAP25(C-to-G) and fed with alkyne-palmitate overnight. Cells were then lysed, and the GFP-tagged constructs were immunoprecipitated. Subsequently, a click reaction with Cy5-azide was performed to label incorporated palmitate, and the samples were subjected to SDS-PAGE and Western blotting. The amount of protein was quantified by immunolabelling (anti-GFP antibody/IRDye-labelled secondary antibody) and used for normalization of the palmitate-Cy5 signal, yielding the palmitate/GFP ratio. The panels show the fluorescence of the palmitate-Cy5 (top) and the GFP signal (bottom) of one representative immunoblot. The bar chart shows palmitate/GFP ratios of n = 5 independent experiments (mean + S.E.M.; t-test *p<0.05, **p<0.01, ***p<0.001, ns = not significant).

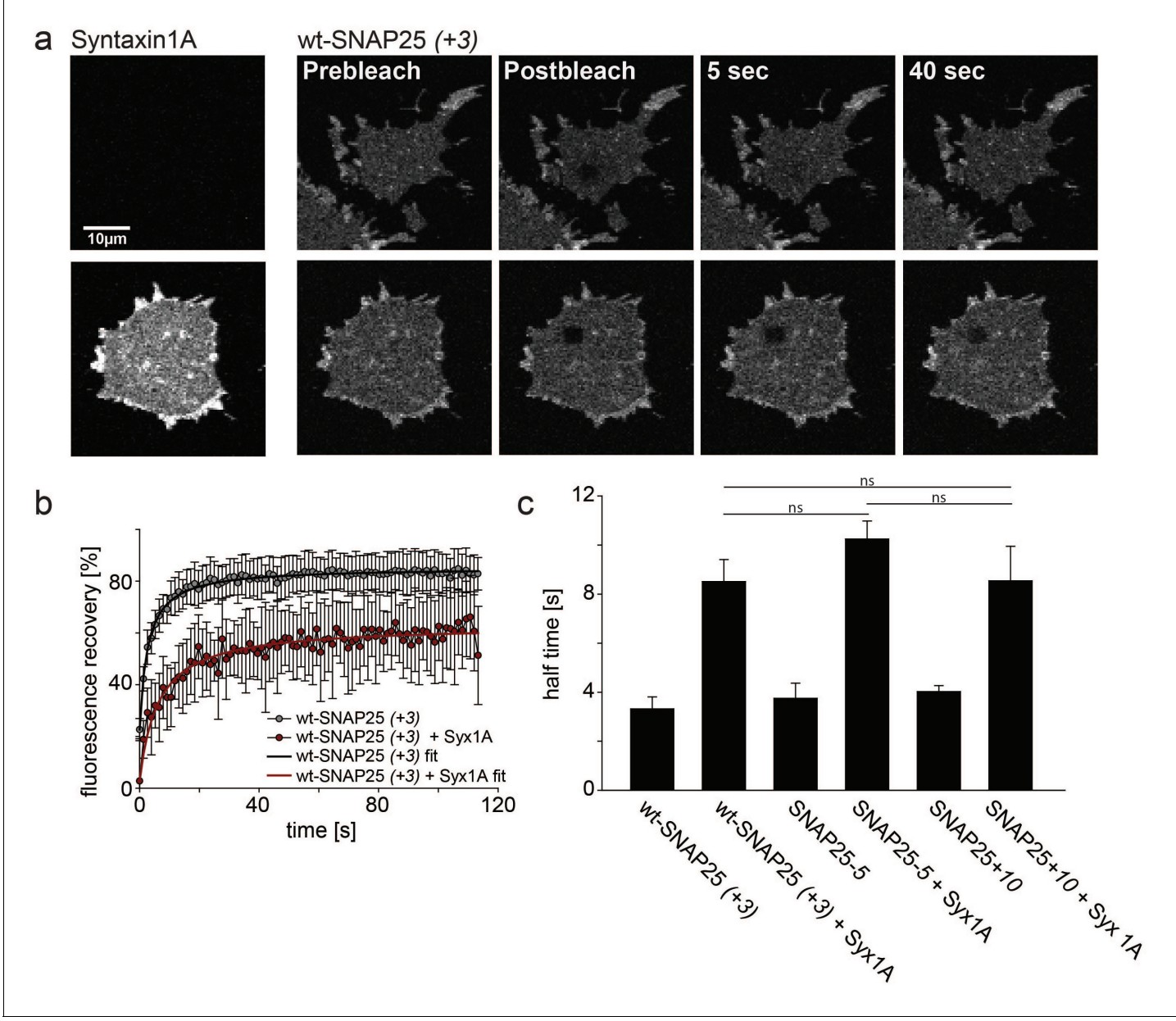

**Figure 6.** Probing the interaction of SNAP25 constructs with syntaxin 1A. (**a**) Illustration of a fluorescence recovery after photobleaching (FRAP) experiment that measures SNAP25 interactions with syntaxin on membrane sheets generated from PC12 cells expressing the respective constructs. GFP-SNAP25 mobility was analysed in the absence (top row) or presence (bottom row) of co-expressed syntaxin 1A-RFP (for RFP fluorescence see images on the left shown at the same scaling). Right (from left to right): membrane sheets before bleaching of a square region of interest (ROI), the first image immediately after bleaching, and 5 s and 40 s after bleaching. The ROI refills with GFP-signal faster in the absence of syntaxin 1A-RFP. (**b**) Averaged fluorescence recovery traces from one experiment, in the absence (grey) or presence (red) of syntaxin 1A-RFP. Values are given as means ± S. D. (n = 7–12 membrane sheets). Hyperbola functions are fitted to the averaged traces yielding the half time of recovery. (**c**) Average half times of recovery for wt-SNAP25 (*+3*), SNAP25-*5* and SNAP25+*10*, in the absence and presence of overexpressed syntaxin 1A. Values are given as means ± S.E. M. (n = 3–4; t-test *p<0.05, **p<0.01, ***p<0.001, ns = not significant). Please note that, in this experiment, large pixels were used to keep bleaching low. Therefore the spatial resolution is lower than that in the other experiments and does not allow for resolving the SNAP25 micropatterning.

## Binding of SNAP25 constructs to liposomes

Next, we tested whether membrane association of SNAP25 is directly mediated by negatively charged lipids. We used reconstituted liposomes containing distinct lipid compositions but lacking

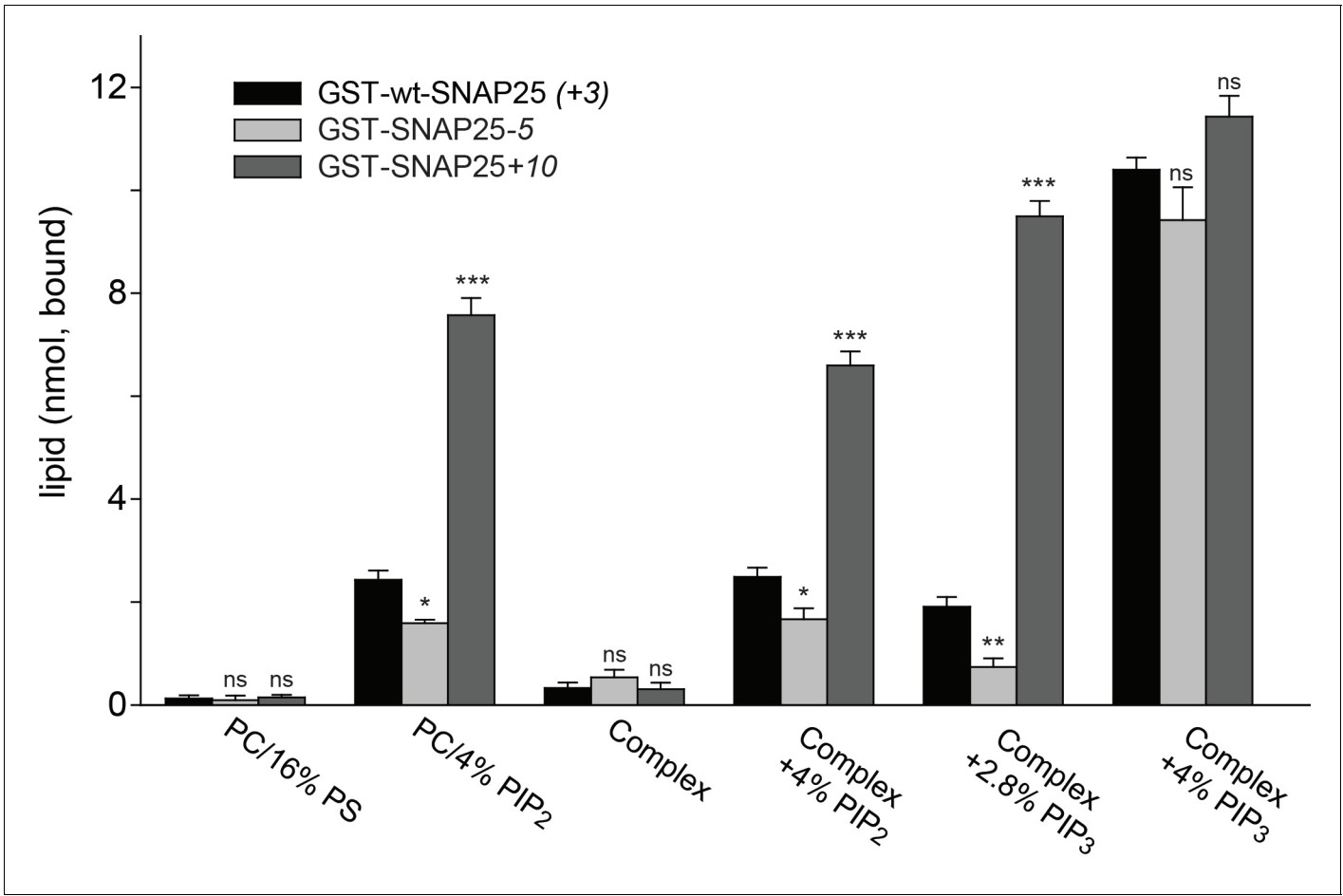

**Figure 7.** Binding of purified SNAP25 constructs to reconstituted liposomes. Atto647N-labeled liposomes containing either POPC or a complex lipid mixture (POPC/DOPS/POPE/cholesterol/PI) and the indicated amounts of $PI(4,5)P_2$ or $PI(3,4,5)P_3$ were added to immobilized GST-wtSNAP25 (+3), GST-SNAP25-5 and GST-SNAP25+10 and incubated 1 hr at 4°C. After washing the beads, the amount of bound liposomes was measured by their Atto647N fluorescence (excitation: 639 nm, emission: 669 nm). The amounts of liposomes specifically bound to the different GST-SNAP25 constructs were determined by subtracting the values derived from the GST controls. Values are given as means ± S.E.M. (n = 3; t-test *p<0.05, **p<0.01, ***p<0.001, ns = not significant).

any proteins (thus eliminating the role of potential SNAP25 binding partners such as syntaxin 1) (*Figure 7*). This assay also clarifies whether the few amino acids located in the relatively small SNAP25 segment do indeed affect lipid binding. GST-tagged wt-SNAP25 (+3), SNAP25-5 and SNAP25+10 were expressed in bacteria and thus lack the palmitoylation modification. The constructs were purified, immobilized on glutathione beads and incubated with Atto647N-PE-labelled liposomes containing either PC/PS or PC/PS/PE/PI/cholesterol in the absence or the presence of distinct phosphoinositides. SNAP25-bound liposomes were quantified by their Atto647N-PE fluorescence and the results were corrected for unspecific binding to GST-beads. All SNAP25 constructs show only weak liposome binding in the absence of phosphoinositides, which excludes phosphatidylserine (PS) as a major/sole binding partner (*Figure 7*).

The presence of 4% $PI(4,5)P_2$ significantly enhanced SNAP25 binding in the PC samples and in the complex lipid mixture. As in the cellular assays, the reduction or addition of positive charges reduced or increased the SNAP25 interaction, respectively. Replacing $PI(4,5)P_2$ by $PI(3,4,5)P_3$ profoundly increased binding, but largely diminished the differences between SNAP25 constructs. This suggests that the presence of additional negative charges creates additional binding contacts, which are probably outside of the cysteine-rich region. In order to keep the number of negative charges comparable to those provided by 4% $PI(4,5)P_2$, $PI(3,4,5)P_3$ was reduced to 2.8%. This condition

yielded a stronger difference in binding in the range between the cell fractionation (*Figure 3*) and the membrane sheet assay (*Figure 4*). Although the magnitudes of the effects are difficult to compare because of the different assay systems (the complex composition of the plasma membrane versus the simple lipid mix in liposomes, concentrations of binding partners, and the presence of SNAP25-palmitoylation in the cellular experiments may modulate the outcome), the observations point to primary interactions that occur in the cysteine-rich region.

Since PS is not capable of recruiting SNAP25, and as $PI(3,4,5)P_3$ is much less abundant in the cellular membrane than $PI(4,5)P_2$ (*Balla, 2013*), these findings point to $PI(4,5)P_2$ as the most likely intracellular binding partner. However, the effect of $PI(3,4,5)P_3$ on wt-SNAP25 (*+3*) binding was very prominent. Thus, membrane microdomains that are locally enriched in $PI(3,4,5)P_3$ at the plasma membrane or in an endosomal compartment (*Wang and Richards, 2012*) could also be preferential sites for SNAP25 targeting. The liposome binding assay thus identifies phosphoinositides as primary membrane-targeting factors interacting with positive charges in the vicinity of the cysteine cluster. However, we cannot exclude the possibility that other sites in SNAP25 contribute to phosphoinositide-dependent binding.

## Discussion

In conclusion, we suggest that after physically contacting the membrane, a SNAP25/SNAP23 protein increases its dwell time at the plasma membrane through an electrostatic mechanism. The basic residues either directly bind to acidic lipids or produce a local positive electrostatic potential which attracts acidic PIPs. The latter mechanism has been described for MARCKS, where 13 basic residues laterally sequester three $PIP_2$ molecules (*Gambhir et al., 2004*). The increased membrane binding observed after introducing additional positive charges (SNAP25+*10*) may reflect a MARCKS-protein-like $PIP_2$-binding behaviour not physiologically relevant for the targeting of SNAP25/SNAP23. Once formed, such a protein-lipid aggregate seems difficult to dissolve, as SNAP25+*10*$_{(C-to-G)}$ remains attached to the membrane for more than 30 min (*Figure 4*).

Palmitoylated cysteine residues are often preceded and/or followed by basic amino acids (*Bizzozero et al., 2001*). It has been speculated that these positively charged residues bind to the negatively charged acyl-coenzyme A, and thus augment the acylation rate. Our data do not argue against this hypothesis but show that these residues have a distinct function in anchoring the protein to the cell membrane. This is supported by the *in vitro* liposome binding assay, which shows the charge-dependent binding in the absence of palmitoylation (*Figure 7*).

Elimination of positive charges distal from the cysteine cluster has no effect on targeting (*Figure 1f*; construct SNAP*25*(*R191A, R198A, K201A*)). This suggests that a random electrostatic contact is not sufficient for targeting. Rather, electrostatic anchoring needs to position the cysteine (s) in a way that facilitates palmitate attachment. The electrostatic contact thus needs to be established by amino acids close to the cysteine residues. Such a mechanism would also explain why the diminishment of positive charges is always accompanied by a loss of targeting, because fewer positive charges will decrease the binding affinity and/or change the geometry of the established contact. On the other side, not all mutants with an increased positive charge show increased membrane targeting. Some actually exhibit a trend towards lesser binding. Perhaps the contact can also become too tight for the attachment of palmitates. Hence, the short targeting motif seems to be optimized to mediate electrostatic anchoring, while still allowing for access to the cysteines for the attachment of palmitates.

Previous reports on plasma membrane-localized, small GTPases showed that most of them contain two or three polybasic subclusters, each spanning about four to five amino acids (*Heo et al., 2006*). Removal of one subcluster abolished plasma membrane targeting, leading to the proposal that association results from the additive binding energies of individual subclusters (*Heo et al., 2006*). Similarly, the more abundant positive charges in SNAP25+*10*$_{(C-to-G)}$ allow membrane association without palmitoylation.

Why do SNAP25 and SNAP23 carry only a modest excess of positive charges instead of several charged clusters like those of the small GTPases? It is tempting to speculate that the electrostatic force, based on three positive charges in wt-SNAP25 (*+3*), has been optimized for mediating anchoring that is just sufficient to increase the dwell time at the plasma membrane, but weak enough for dissociation after depalmitoylation. Otherwise, electrostatic anchoring would interfere with SNAP25

palmitoylation/depalmitoylation during SNAP25 recycling (see 'Introduction'). Indeed, the SNAP25 $+10_{(C-to-G)}$ construct remains attached to the plasma membrane without being palmitoylated.

Our data also show that hydrophobic forces (*Greaves et al., 2009*) are less crucial than electrostatic contacts in the plasma membrane binding of SNAP25. Although the hydrophobic amino acids in the cysteine cluster remain unchanged, the removal of charges abolishes targeting, even though lysines are exchanged for more hydrophobic residues (alanines in the constructs SNAP25-$1_{distal}$, SNAP25-$1_{proximal}$ and SNAP25-5, and leucines in SNAP25-$5_{hydrophob}$).

Hence, in a cascade of interactions, a modest modulation of the local charge regulates the efficiency of the cellular processes. It seems there are also other membrane contacts that could depend on charge, for instance regulation of the docking time of lipid transfer proteins. Here, the exchange of five lysines for glutamates in the protein loops that interact with the membrane leads to loss of Osh4-mediated sterol transfer (*Schulz et al., 2009*).

In conclusion, we have identified a mechanism of initial plasma membrane association of SNAP25 and SNAP23, which precedes the stable membrane attachment mediated by palmitoylation of cysteines. The mechanism is based on a relatively small cluster of basic amino acid residues that anchor the protein by binding to acidic lipids, in particular to polyphosphorylated phosphoinositides, with $PI(4,5)P_2$ being the most likely candidate. The data suggest that even small changes in protein electrostatics can have strong effects on a cellular mechanism, merely by transiently anchoring a protein to its site of destination.

## Materials and methods

### Plasmids

Plasmids for the expression of GFP-SNAP25 (*Halemani et al., 2010*) and GFP-SNAP23 are based on the expression vector pEGFP-C1 (GenBank accession No. U55763, Clontech, Mountain View, CA), which contains a monomeric variant of mEGFP fused N-terminally to the sequence of full-length rat SNAP25B (NP_112253.1) or SNAP23 (NP_073180). All fusion proteins contain a linker of five amino acids between mEGFP and the N-terminus of SNAP25B or SNAP23. Mutations in the SNAP25 or SNAP23 coding sequence were introduced via fusion PCR with purchased oligonucleotides (Eurofins Genomics), followed by insertion into the above-mentioned expression vector using the SacI and BamHI sites, or in the case of SNAP25$_{(C-to-G)}$, the XhoI and KpnI restriction sites. The construct for expression of C-terminally RFP-tagged rat Syntaxin 1A is based on the expression vector pEGFP-N1 (GenBank accession No. U55762, Clontech, Mountain View, CA), into which Syntaxin-RFP is inserted using the XhoI and the NotI restriction sites. A twelve amino acid linker connects Syntaxin 1A (NP_446240) to a monomeric RFP (*Campbell et al., 2002*) lacking the first amino acid.

For expression of GST-tagged constructs, the coding sequences for wt-SNAP25 (+3), SNAP25-5 and SNAP25+10 were amplified with primers carrying restriction sites for BamHI (forward primer) and EcoRI (reverse primer). The sequences were first subcloned into the pGEM-T easy vector system (catalogue no. A1360, Promega) by TA cloning, and in a second step subcloned into the expression vector pGEX-6P1 (GE Healthcare Life Sciences) via the BamHI and EcoRI restriction sites.

All constructs were verified by sequencing.

### Cell culture and generation of membrane sheets

PC12 cells (a gift from Rolf Heumann, Bochum, Germany; similar to clone 251 (*Heumann et al., 1983*)) were cultured in DMEM with high (4.5 g/l) glucose (PAN biotech) supplemented with 10% horse serum (Biochrom), 5% fetal calf serum (Biochrom) and 100 U/ml penicillin/100 ng/ml streptomycin (PAN biotech). During the course of the project, characteristic features of the cell line — such as morphology, expression of neuronal proteins, and their responsiveness to NGF — were regularly confirmed. Cells were maintained at 37°C and 5% $CO_2$ in a sterile incubator and tested negative for mycoplasmic infections (GATC Biotech, Konstanz, Germany).

Cells were transfected with the Neon Transfection System (Thermo Fisher Scientific, Waltham, MA, USA). The tip (100 μl) was loaded with 10 μg plasmid DNA of each construct. Cells were transfected by applying a pulse at 1410 V and 30 ms pulse width. Cells were plated onto poly-L-lysine (PLL) (Sigma, Cat. No: P-1524) coated coverslips (25 mm diameter, Menzel Gläser, Braunschweig, Germany)) and maintained for at least 48 hr before imaging.

For membrane sheet generation, cells were subjected to a brief ultrasound pulse in ice-cold sonication buffer (120 mM KGlu, 20 mM KAc, 20 mM HEPES-KOH, 10 mM EGTA; pH 7.2).

## Confocal microscopy and analysis

PC12 cells were imaged at 37°C in Ringer solution (130 mM NaCl, 4 mM KCl, 1 mM $CaCl_2$, 1 mM $MgCl_2$, 48 mM D(+)αGlucose, 10 mM HEPES; pH 7.4) using a FluoView1000 confocal laser scanning microscope (Olympus) and a UPLSAPO 60x oil objective (NA 1.35). Focussing on the glass–cell interface, all green cells were imaged provided they displayed a clear cell–glass contact area. This excludes dead or dying cells that are in the process of rounding up and detaching. In addition, we avoided large clumps of cells because for such cells it is difficult to identify a longer section of cell membrane required for analysis. For measuring membrane-to-cytosol ratios, PC12 cells expressing GFP-tagged constructs were scanned in a 256 pixel x 256 pixel field (12 bit image) at a pixel size of 137 nm using for excitation 488 nm. The equatorial plane of the cell was imaged. To facilitate subsequent linescan analysis, the scanned field was rotated to allow for each cell placing a horizontal linescan (length 100 pixel; five pixel width, averaged) perpendicular to the plasma membrane that records the averaged fluorescence intensities along the 100 pixel linescan (including background fluorescence). Fluorescence intensities were background corrected and normalized to the peak intensity at the plasma membrane. Traces from several cells imaged that day (ranging from 14–40 per condition) were aligned with reference to the peak intensity at the plasma membrane. The cytosolic fluorescence level was averaged over five pixels starting at a 10 pixel distance from the peak intensity at the plasma membrane, and the cell periphery/cytosol ratio was calculated subsequently.

For mobility measurements by FRAP, the laser intensities of the 488 nm laser (for GFP) and the 543 nm (for RFP) were reduced to a minimum to prevent bleaching effects during the recording. Membrane sheets were analysed by scanning a 100 pixel x 100 pixel field with a pixel size of 0.414 µm. Recordings started with a pre-bleaching phase of three images, followed by a 500 ms bleaching step and the recording of the recovery phase. For bleaching, a region of interest (ROI) with a size of 7 pixels x 7 pixels (2.9 µm x 2.9 µm) was bleached using a 488 nm laser in combination with a 405 nm laser (both set to their maximum intensity). After bleaching, image sequences were taken at 1.2 Hz for 113 s with the scanning speed set to 40 µs per pixel. Recovery traces were background-subtracted and normalized to the average of the pre-bleach values. For one experiment, several normalized recovery traces were averaged (7–15 membrane sheets per condition). A hyperbolic curve $y(t) =$ offset + maximal recovery x $t/(t + t_{1/2})$ was fitted to the averaged recovery trace, yielding the half time ($t_{1/2}$) of recovery.

## Epi fluorescence microscopy

To measure the association of SNAP25 constructs with the plasma membrane on plasma membrane sheets, a Zeiss Axio Observer D1 epifluorescence microscope equipped with a Plan-Apochromat 100x/NA 1.4 oil immersion objective and a 12 bit CCD camera (1376 × 1040 pixel) was used, yielding a pixel size of 64.5 nm x 64.5 nm. Freshly prepared membrane sheets were imaged in sonication buffer supplemented with TMA-DPH (1-(4-tri-methyl-ammonium-phenyl)−6-phenyl-1,3,5-hexatriene-p-toluenesulfonate; Thermo Fisher Scientific), up to ≈ 35 min after membrane-sheet generation. TMA-DPH staining was applied for visualization of the shape and integrity of the membrane sheets. Pictures were taken using filter sets F11-000 (AHF Analysentechnik, Tübingen, Germany) for TMA-DPH (blue channel) and F36-525 (AHF Analysentechnik, Tübingen, Germany) for GFP (green channel). From individual membrane sheets, the fluorescence intensity was measured in 30 pixel x 30 pixel ROIs and background subtracted. For each experiment and condition, the values of 21–96 membrane sheets were averaged.

## Membrane fractionation and western blot

About 48 hr after transfection, $9 \times 10^6$ PC12 cells were detached from the substrate by trypsin treatment for 2 min (0.05% Trypsin and 0.02% EDTA in PBS, PAN Biotech, Cat# P10-0231SP) and trypsin was inactivated by adding medium. Cells were pelleted by centrifugation at 1000 x g at room temperature for 3 min and the pellet was washed with PBS. Cell pellets were resuspended in 750 µl ice-cold homogenization buffer (300 mM sucrose, 5 mM Tris-HCl, 0.1 mM EDTA, 1 mM PMSF freshly added; pH 7.4). Using a Potter-Elvehjem homogenizer, cells were kept on ice and homogenized in a

volume of 0.75 ml by applying 100 strokes. The homogenate was centrifuged for 8 min at 800 x g at 4°C, yielding pellet P1 (containing non-homogenized cell debris) and supernatant S1. S1 was transferred into a new tube for a second centrifugation step for 120 min at 20,000 x g and 4°C, yielding pellet P2 (containing the enriched membrane fraction, which was resuspended in 750 μl homogenization buffer) and the supernatant S2 with the cytosolic fraction. Protein concentrations of P2/S2 were adjusted to the lowest concentration in the series, before adding respective amounts of 4x Laemmli buffer followed by incubation at 95°C for 10 min. Samples of 10 μg per lane were subjected to SDS-PAGE analysis using a 12% polyacrylamide gel. Proteins were then transferred onto a Roti-NC nitrocellulose membrane (Carl Roth, Germany) applying semi-dry blotting. Nitrocellulose membranes were blocked with PBS-T (0.05% Tween20 in PBS) containing 5% milk powder for one hour, and incubated with primary antibody against GFP diluted in blocking solution overnight at 4°C. Membranes were washed for 20 min with PBS-T three times. For detection a second antibody tagged with HRP (RRID:AB_631747, Cat# sc-2030, Santa Cruz Biotechnology, USA) was applied for 1 hr and, after washing three times with PBS-T, chemiluminescence was detected with Luminol Reagent (sc-2048, Santa Cruz Biotechnology, USA) using autoradiographic films. Films were scanned and band intensities were quantified from the digital images.

## Assessment of SNAP25 palmitoylation and western blot

Ten million PC12 cells were transfected with 15 μg GFP fusion constructs of wt-SNAP25 (+3), SNAP25-5, SNAP25+10 or SNAP25$_{(C-to-G)}$. After one hour, the medium was replaced with DMEM containing 15% delipidized FCS (PAN biotech) and 100 μM palmitate-alkyne (a kind gift from the Thiele lab, LIMES Institute, Bonn). After 15 hr of feeding, cells were harvested via trypsinization and scraping, washed once with PBS, and resuspended in lysis buffer (1% Triton, 1x cOmplete protease inhibitor cocktail, 150 mM NaCl, 5 mM MgCl$_2$, 25 mM HEPES; pH 7.2). Lysis was promoted by vortexing and sonication. Samples were then centrifuged for 10 min at 14,000 x g, and the supernatant was bound to a GFP-trap (Chromotek) for 2 hr at 4°C to immunoprecipitate the GFP-SNAP25 fusion constructs. After several washing steps, incorporated palmitate alkyne was clicked to a Cy5-labelled azide (Sigma, c$_{final}$ = 100 μM) in 100 mM HEPES, pH 7.2, containing 500 μM tetrakis(acetonitrile)copper(I)tetrafluoroborate (Sigma) for 1 hr at 37°C. The samples were then washed to remove non-bound Cy5, and the GFP-constructs were eluted by boiling in Laemmli buffer, and loaded onto an SDS-PA gel. Proteins were then wet blotted to a nitrocellulose membrane. The membrane was blocked with a 1:1 mixture of Odyssey blocking buffer (Li-Cor) and PBS, and then incubated with a rabbit anti-GFP antibody (RRID: AB_303395; abcam, catalog no. ab-290, diluted 1:1000 in blocking solution containing 0.1% Tween-20). After washing, the membrane was incubated with an IRDye 800CW-coupled goat-anti rabbit secondary antibody (Li-Cor, catalog no. 9263221, diluted 1:10,000 in blocking solution containing 0.1% Tween-20). The Cy5 fluorescence of the palmitate and the IRDye 800CW fluorescence of the GFP were imaged with an Odyssey infrared imaging system (Li-Cor) at 700 nm and 800 nm, respectively. The fluorescent bands were quantified using ImageJ's Gel Analyser.

## Liposome preparation

Atto647N-DPPE (Att647N-1,2-dipalmitoyl-sn-glycero-3-phosphoethanolamine) was purchased from Atto-Tec. All other lipids were from Avanti Polar Lipids. The complex lipid mixture (5 μmol total amount of lipid) contains 34.5 mol % 1-palmitoyl-2-oleoyl-sn-glycero-3-phosphocholine (POPC), 15 mol % 1,2-dioleoyl-sn-glycero-3-phosphoserine (DOPS), 20 mol % 1-hexadecanoyl-2-octadecenoyl-sn-glycero-3-phosphoethanolamine (POPE), 25 mol % cholesterol (from ovine wool), 5 mol % liver L-α-phosphatidylinositol (PI, from liver) and 0.5 mol % Atto647N-DPPE. For the lipid mixes containing brain L-α-phosphatidylinositol-4,5-bisphosphate (PI(4,5)P$_2$) or 1-stearoyl-2-arachidonoyl-sn-glycero-3-phospho-(1'-myo-inositol-3',4',5'-trisphosphate) (PI(3,4,5)P$_3$) the amount of PI was reduced accordingly. The lipids were dissolved in chloroform or chloroform/methanol (3:1 ratio, for PI(4,5)P$_2$ and PI(3,4,5)P$_3$), mixed and dried under a flow of nitrogen. The remaining chloroform was removed by vacuum for 4 hr. The lipids were dissolved in 1 ml reconstitution buffer (25 mM HEPES/KOH pH 7.4, 200 mM KCl, 1% (w/v) OG (n-Octyl-$\beta$-D glucopyranoside), 1 mM DTT (1,4-dithiothreitol)) by 30 min shaking. To form liposomes, OG was diluted below the critical micelle concentration by the addition of 2 ml buffer (25 mM HEPES/KOH; pH 7.4, 200 mM KCl, 1 mM DTT). The residual OG was

removed by flow dialyses with 4 L 25 mM HEPES/KOH pH 7.4, 135 mM KCl, 1 mM DTT overnight. Subsequently, a Nycodenz gradient centrifugation was performed to isolate the liposomes. There-fore, the dialyzed samples were mixed with an equal volume of 80% (w/v) Nycodenz and transferred into two SW60-tubes (Beckman Coulter). Layers of 750 µl 35% (w/v) Nycodenz, 150 µl 11.6% (w/v) Nycodenz and 100 µl fusion buffer were added on top of the 40% (w/v) Nycodenz/liposome solution. The gradient was spun at 55,000 rpm for 3 hr 40 min at 4°C. The liposomes were isolated, followed by a buffer exchange (25 mM HEPES/KOH pH 7.4, 135 mM KCl, 1 mM DTT, 0.1 mM EGTA, 0.5 mM $MgCl_2$, 1 mM DDT) using a PD MiniTrap G-25 (GE Healthcare Life Sciences). The amounts of lipids were quantified by measurement of Atto647N fluorescence (excitation 639 nm, emission 669 nm) in a Fluoroskan Ascent FL Microplate Fluorometer (Thermo Scientific).

### Liposome-binding studies

Wt-SNAP25 (+3), SNAP25-5 and SNAP25+10 pGEX-6P1 constructs were transformed into *Escherichia coli* (Rosetta (DE3)pLysS) and expression of the GST fusion proteins was induced with 1 mM IPTG. After harvesting, cells were lysed for 60 min at 4°C in 150 mM NaCl, 50 mM Tris-HCl, pH 7.4 and 1 mM EDTA containing Roche cOmplete protease inhibitor, 1 mM DTT, 100 µg/ml lysozyme and two units/ml DNAse I. The suspension was sonicated, centrifuged for 30 min at 20,000 x g, fro-zen in liquid nitrogen and thawed for binding of the constructs directly to glutathione beads.

To monitor SNAP25 interaction with liposomes, 42 µg GST-SNAP25 constructs or a equimolar amount of GST (glutathione S-transferase) were bound to 20 µl glutathione (GSH) sepharose four fast flow beads (GE Healthcare Life Sciences) prewashed 3 x with $ddH_2O$ and 3 x with fusion buffer (25 mM HEPES/KOH pH 7.4, 135 mM KCl, 1 mM DTT, 0.1 mM EGTA, 0.5 mM $MgCl_2$, 1 mM DDT). 160 nmol liposomes in fusion buffer were added to the beads and incubated 1 hr at 4°C on a rota-tion wheel. The beads were washed once with 1 ml fusion buffer and resuspended in 80 µl fusion buffer. The bound liposomes were detected by measuring the Atto647N fluorescence. The amounts of liposomes specifically bound to the different GST-SNAP25 constructs were calculated by subtract-ing the values derived from the GST controls.

## Acknowledgements

The authors would like to thank Michael Pankratz and Nora Karnowski for comments on the manu-script. This work was supported by a grant from the Deutsche Forschungsgemeinschaft (TRR83 to THS and TL).

## Additional information

### Funding

| Funder | Grant reference number | Author |
| --- | --- | --- |
| Deutsche Forschungsge-meinschaft | TRR83 | Thomas H Söllner<br>Thorsten Lang |

The funders had no role in study design, data collection and interpretation, or the decision to submit the work for publication.

### Author contributions

PW, Conceptualization, Formal analysis, Supervision, Investigation, Writing—original draft, Writing—review and editing; HB, Conceptualization, Formal analysis, Supervision, Investigation, Writing—review and editing; KMR, Conceptualization, Formal analysis, Investigation; SD, KP, Formal analysis, Investigation; THS, Conceptualization, Supervision, Funding acquisition, Writing—review and edit-ing; TL, Conceptualization, Supervision, Funding acquisition, Writing—original draft, Project adminis-tration, Writing—review and editing

### Author ORCIDs

Thorsten Lang, http://orcid.org/0000-0002-9128-0137

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
