## [Decision Letter]

[Editors’ note: this article was originally rejected after discussions between the reviewers, but the authors were invited to resubmit after an appeal against the decision.]

Thank you for submitting your work entitled "Electrostatic anchoring precedes stable membrane attachment of SNARE proteins with the plasma membrane" for consideration by *eLife*. Your article has been favorably evaluated by Randy Schekman (Senior Editor) and three reviewers, one of whom is a member of our Board of Reviewing Editors. The reviewers have opted to remain anonymous.

Our decision has been reached after consultation between the reviewers. Based on these discussions and the individual reviews below, we regret to inform you that your work will not be considered further for publication in *eLife* in the current form. Although the reviewers and editors found the topic of your work of considerable interest for potential publication in *eLife*, serious concerns were raised that may require a substantial amount of additional experiments and computer simulations.

Reviewer #1:

This work draws attention to an aspect of SNARE proteins that has not been extensively studied. Basic residues were identified in the cys cluster of SNAP25/SNAP23 that promotes initial plasma membrane association of SNAP25/SNAP23 which precedes the stable membrane attachment mediated by palmitoylation of cysteines. The authors performed an extensive set of mutagenesis, imaging, and fractionation experiments to support this conclusion. Coarse grain MD simulations were then performed to study random collisions between the corresponding peptides and a membrane and conclude that PIP2 is essential for the membrane association of the peptide. They support this notion by a competition experiment between SNAP-25 and the PH domain of phospholipase C-δ for PIP2.

1) In Figure 1, the ratio of membrane/cytosol localization is calculated. However, comparing the absolute protein level that is associated with the membrane would also be informative, considering that high expression levels of the positively charged mutants might cause saturation of membrane binding.

2) In Figure 2, the total expression level of SNAP-25 should be provided in addition to the cellular fractions.

3) In Figure 4, images should be provided to correspond to all cases shown in the bar chart.

4) In Figure 5 negative control should be provided that shows that the mobility of SNAP-25 is indeed affected by interaction with syntaxin, e.g., by introduction of mutations in SNAP-25 that interfere with SNARE complex formation.

5) In Figure 8, a control should be provided to show that the competition is related to the competing interaction between the PH domain and PIP2, e.g., by using a mutant of the PH domain that does not interact with PIP2.

6) The role of polybasic residues near palmitylation or myristoylation sites has been reported previously in other contexts, and would be useful to provide a brief summary in the Introduction, e.g., M. Crouthamel, et al. Cell Signal, 20 (2008), pp. 1900-1910; K.H. Pedone, J.R. Hepler. J Biol Chem, 282 (2007), pp. 25199-25212; K.A. Cadwallader, et al. Mol Cell Biol, 14 (1994), 4722-4730; Wright, L. P. & Philips, M. R. J. Lipid Res. 47, 883-891 (2006); O. Jeffries, et al. J Biol Chem, 287 (2012), 1468-1477.

7) There are a few polybasic amino acids in the C-terminal part of the of SNAP-25 linker, such as R191, R198 and K20 that may also contribute to plasma membrane localization. An experiment would be optional, but at the minimum, the authors should comment on these residues.

Reviewer #2:

This is an interesting work dealing with the determinants of membrane association and subsequent palmitoylation in SNARE proteins, with potential generality. The plasma membrane localization data and how they respond to mutation are compelling to this reviewer. The simulation methodology is not clearly specified in several key respects, and the stable association (+/1 1 peptide) of all peptide sequences tested raises questions about the simulation model and its ability to capture the desired behavior. Time until stable association is not a measure of equilibrium properties and is thus inappropriate as a metric. The equivalent metric to the experimental data is an estimate of equilibrium partition coefficient.

Since peptide secondary structure (and 3D structure also) will differ between solution and membrane-associated forms, simply constraining structure and measuring the association (or even partition) between solution and membrane-associated forms does not capture either the kinetics or equilibrium behavior of the adsorption process. Atomistic simulations of the peptides in membrane-bound and solution forms to measure structural equilibria would be required to complete this analysis.

The authors state that they used PME electrostatics with MARTINI, but they do not state whether they also used the MARTINI polarizable water model, which is required for proper usage of PME electrostatics in the model as per the original papers. This technical point is important here, as the authors are measuring electrostatic interactions between charged peptides and a membrane (and they appear to observe artifactually stable association).

In view of these issues, I would recommend that the simulations either be redone entirely or eliminated from the manuscript, as they do not provide a robust measure of the phenomena the authors are trying to predict (and indeed measure experimentally).

Reviewer #3:

SNAP25 and SNAP23 are lipid anchored to membrane by post-translational palmitoylation of cysteine residues. The authors present evidence that initial access of SNAP25 and SNAP23 onto membrane for subsequent palmitoylation is mediated by electrostatic interactions of Lys residues near the Cys quartet with acidic phospholipids.

1) The work seems preliminary in only evaluating two SNAP25 constructs, one removing 8 Lys (SN25-5) and the other adding 4 Lys (SN25+10). The largest effects were observed with the first of these so it would be of interest to test other mutants to determine proximity to Cys quartet. Many Lys residues have acidic neighbors and it is unclear whether replacement of the Asp70, Glu73,-75 and Asp80 would be equally disruptive. The work on SN25+10 is not very relevant because it does not deal with requirements in the native protein.

2) Others have suggested that hydrophobic residues are involved in targeting but this issue is not addressed or much discussed. Would replacing the same residues with hydrophobic residues preserve membrane targeting?

3) The conclusions of the article would be much stronger if an assessment of palmitoylation were conducted. The authors instead infer palmitoylation based on Cys to Gly.

4) Data are shown as SEM but there is no statistical analysis for Figure 1, Figure 2, Figure 3, Figure 4, Figure 5, Figure 6, Figure 8 so it is unclear what differences are significant.

5) The results shown in Figure 8 are quite puzzling. The images shown do not seem to be reflective of the intensity differences plotted in Figure 8. The left panels of 8A show a remarkable suppression of PLCPH by SN25 (but not by C to G) whereas histograms indicate 2X change. It is not clear which of these intensity differences is statistically significant. There is no indication that similar amounts of protein were being expressed in these studies in comparisons done in a cell line that has endogenous SN25. Lastly, it is not at all clear what the interpretation of these studies would be.

6) The images of the membrane sheets (Figure 4) are confusingly shown at different scales and cannot be compared to bar graphs. These should be shown as direct comparisons to agree with the bar graphs.

7) To eliminate expression level as an important variable in Figure 1, the authors provide Figure 1—figure supplement 1 to indicate lack of correlation. However, the ordinate scale greatly exceeds the range over which the results of Figure 1 were taken so the lack of correlation over this broad scale did not seem to be tailored to the experimental work shown.

8) Two features of Figure 5 need to be addressed. Firstly, the red and green channels exhibit very close similarity in the lower panels. Most previous studies on Syx/SN25 do not show this degree of alignment so the possibility of channel overlap should be addressed. Secondly, overexpression of Syx appeared to slow the mobility of all fluorescent objects in the lower row compared to upper row. Is the result of extreme overexpression?

[Editors’ note: what now follows is the decision letter after the authors submitted for further consideration.]

Thank you for resubmitting your work entitled "Electrostatic anchoring precedes stable membrane attachment of SNAP25/SNAP23 to the plasma membrane" for further consideration at *eLife*. Your revised article has been favorably evaluated by Randy Schekman (Senior Editor), a Reviewing Editor, and two reviewers.

The manuscript has been improved but there are two remaining issues that need to be addressed before acceptance, as outlined below:

1) To reinforce the notion that SN25 membrane translocation requires PIP2, the authors construct the study of Figure 8 of expressing the PH domain of PLCdelta1 and finding less SN25 on membrane sheets. However, in this study the PH domain was expressed for 48 hrs, there was less SN25 expressed (compared to PHmut), and the cells are drastically altered in their morphology. The latter suggests drastic cytoskeletal changes or apoptosis that occur in depriving a cell long term of PIP2. Either of these would undoubtedly affect SN25 trafficking through an endosomal pool and its turnover in the cells. There does not seem to be a straightforward interpretation for the results of Figure 8. In the absence of more concerted characterization, this study should be removed from the manuscript.

2) In Figure 7, in vitro protein-liposome interaction studies show that WT SN25 binding to liposomes is enhanced by inclusion of PIP2. The SN25-5 mutant exhibits some decrease in binding. This could be interpreted as PIP2 playing some role in recruiting SN25 to the membrane although the loss of in vitro binding by SN25-5 is much smaller than that claimed for cellular membrane interactions. This difference between the in vitro binding studies and the cellular studies should be discussed. Also, the results with SN25+10 do not seem to be relevant because this mutant has acquired additional MARCKS protein-like properties of PIP2 binding that may not be relevant to the WT SN25 protein.

---

## [Author Response]

[Editors’ note: the author responses to the first round of peer review follow.]

Please note that after adding more constructs we changed the constructs’ naming to avoid confusion. Now, the nomenclature is SNAP25*x* or SNAP23*x*, with *x* indicating the net charge of the cysteine rich region (for example see revised Figure 1; in the schemes we also changed the colour scheme a little to make it more suitable for red-green blinds). A subscript is added in case there are two constructs with the same net charge. Hence, the constructs which were previously named SN25mut-5 and SN25mut+10 are now referred to as SNAP25*-5* and SNAP25*+10*.

Reviewer #1:

[…] 1) In Figure 1, the ratio of membrane/cytosol localization is calculated. However, comparing the absolute protein level that is associated with the membrane would also be informative.

In Figure 8; left panel, we show the absolute protein fluorescence which originates from the cell membrane. In this plot, a loss in membrane-binding is observed for SNAP25*- 5*, but there is no increase in membrane association for SNAP25*+10.* This is due to the fact that expression levels of SNAP25*+10* tended to be lower in this set of experiments. When “correcting” the membrane fluorescence for the expression level, the membrane fraction of SNAP25*+10* is found to be higher than for wt-SNAP25 *(+3)* (Figure 8; right panel). In this latter depiction, the membrane association of the three constructs is similar to the distribution observed in membrane sheets (Figure 4).

Author response image 1.From a set of experiments shown in Figure 1 the difference “periphery” – “cytosol” = “membrane” was calculated (left), and normalized to the “cytosol”-value (right).Values are given as means ± s.e.m. (n = 4 independent experiments).**DOI:**
http://dx.doi.org/10.7554/eLife.19394.015

However, this correction is flawed: in the line scan analysis, the value obtained at the cell periphery is the additive signal of the membrane (which is only 5 nm thick and accounts for about 1% of the cell volume) and the cytosol (which occupies a much larger volume). For the mostly cytosolic SNAP25*-5*, the cytosolic fluorescence might be a proper approximation of the expression level. However, the stronger the construct is membrane-associated the larger the error of this approximation. Therefore, we would prefer not to show the image.

Considering that high expression levels of the positively charged mutants might cause saturation of membrane binding.

If this were the case, the ratio of the periphery-to-cytosol fluorescence should decrease with increasing peak intensities in the line scans. This, however, was not observed (see modified old Figure 1—figure supplement 1, now new Figure 1—figure supplement 2).

This suggests there are no saturation effects of the positively charged mutant. We assume this was difficult to see in the old Figure 1—figure supplement 1 because of the y-axes scaling (see also our reply to issue #7 raised by referee #3). We now use interrupted y-axes so that the distribution of data points between ratios 0 and 15 can be more easily discerned.

2) In Figure 2, the total expression level of SNAP-25 should be provided in addition to the cellular fractions.

We added Figure 3—figure supplement 2 (please note that the old Figure 2 is now Figure 3) which shows the expression level of the SNAP25 constructs. While expression is highly variable between experiments it does not seem that one particular construct tends to be expressed stronger or weaker.

Moreover, we did not note a correlation between expression level and membrane association. This matches the observations we made throughout all imaging experiments.

3) In Figure 4, images should be provided to correspond to all cases shown in the bar chart.

As requested, we added an image for SNAP25*+10* (see revised Figure 6). Regarding SNAP25_(C-to-G)_, we cannot show an image of a membrane sheet of which we are positive that it was produced from a parent transfected cell. To explain this issue, we stated in the Methods section of the old: “During imaging, the sample was screened for green fluorescence and all membranes exhibiting green fluorescence were imaged in the green channel, followed by imaging of the blue channel. In the case of SNAP25_(C-to-G)_, no green fluorescence was visually detectable during the screening process. Therefore, membranes were identified in the blue channel, for quantification of the background signal in the green channel.” We now state in the figure legend of Figure 6 (formerly Figure 4) “During imaging, the sample was screened for green fluorescence and all membranes exhibiting green fluorescence were imaged in the green channel, followed by imaging of the blue channel. We also analysed sheets from cells transfected with SNAP25(C-to-G), but in these samples there was no green fluorescence visually detectable in the screening process”. We thus decided to remove SNAP25_(C-to-G)_ from the bar chart.

To address comment #6 from referee #3, we applied the same lookup table (LUT) to all constructs. However, as the dynamic range in the data is higher than the display dynamics of the screen, we now show two image panels: one for evaluating the pattern of bright fluorescence (at a LUT at which dim membranes appear essentially “black”) and one for evaluating the pattern of the dim fluorescence (with “saturated” bright membranes).

4) In Figure 5 negative control should be provided that shows that the mobility of SNAP-25 is indeed affected by interaction with syntaxin, e.g., by introduction of mutations in SNAP-25 that interfere with SNARE complex formation.

When we published this assay for the first time (Halemani et al., Traffic, 2010; cited in the manuscript), we clarified that the mobility of SNAP25 is indeed affected by interactions with syntaxin by introducing mutations into the SNARE-motifs. For SNARE-complex formation the SNARE-motifs adopt a helical conformation which is disrupted upon insertion of prolines, which in turn affect interactions with syntaxin. We found that the mobility of SNAP25 carrying mutations in the N-terminal part of the first SNARE-motif (Mutations M32P and V36P) is almost independent from the syntaxin concentration. After introducing the mutations in the C-terminal half of the first SNARE- motif (I60P, M64P), which still allows the N-terminal part to partially interact with syntaxin, a 30% slow-down is observed. When removing both SNARE-motifs, the construct diffuses completely independent from the syntaxin concentration. We now state in the manuscript “The slow- down is dependent on a syntaxin-SNAP25 interaction, as it requires the N-terminal SNARE motif of SNAP25. Moreover, a mutant carrying introduced prolines in the N-terminal SNARE-motif of SNAP25 (prolines interfere with the α-helix formation which in turn is required for SNARE- complex formation) moves almost independently of the syntaxin concentration (Halemani et al., 2010).”

5) In Figure 8, a control should be provided to show that the competition is related to the competing interaction between the PH domain and PIP2, e.g., by using a mutant of the PH domain that does not interact with PIP2.

We repeated this experiment in a cell line lacking endogenous SNAP25, as suggested by referee #3, and included a mutant of the PH domain (PH-PLCδ carrying the mutations K32A, W36N and R38K; Flesch et al., Biochem. J., 2005) as a control. In sheets co-expressing SNAP25 and the wildtype PH domain we find 54% less membrane-bound SNAP25 compared to sheets co-expressing the mutant- PH domain.

As suggested by referee #3, we additionally determined the constructs’ expression levels in these experiments. We find a 25% reduction in SNAP25 levels if the protein is co-expressed with wildtype PH-PLCδ compared to mutant PH-PLCδ. When correcting the membrane-associated fraction for the expression level, we still find a 38% reduction in SNAP25 binding in sheets co-expressing wt-PLCδ. We thus state “A role of plasma membrane PIP2 in SNAP25 association is also supported by cellular experiments: cells co-expressing the PH domain of phospholipase Cδ, which has a high affinity for PI(4,5)P_2_ (Stauffer et al., 1998), reduces SNAP25 membrane targeting (Figure 8).” Please see also new Figure 8, new Figure 8—figure supplement 1 and new Figure 8—figure supplement 2.

To provide further evidence for the importance of phosphoinositides, we additionally performed binding studies with liposomes (new Figure 7). The data show that PIP_2/3_ mediate SNAP25 recruitment, and that recruitment efficiency decreases in the order of SNAP25*+10 >* wt-SNAP25 (*+3*) > SNAP25*-5* (for more details please see our reply to referee #2 below).

6) The role of polybasic residues near palmitylation or myristoylation sites has been reported previously in other contexts, and would be useful to provide a brief summary in the Introduction, e.g., M. Crouthamel, et al. Cell Signal, 20 (2008), pp. 1900-1910; K.H. Pedone, J.R. Hepler. J Biol Chem, 282 (2007), pp. 25199-25212; K.A. Cadwallader, et al. Mol Cell Biol, 14 (1994), 4722-4730; Wright, L. P. & Philips, M. R. J. Lipid Res. 47, 883-891 (2006); O. Jeffries, et al. J Biol Chem, 287 (2012), 1468-1477.

We included these references in the Introduction of the revised manuscript after “Polybasic amino acid patches mediate non-specific interactions with anionic lipids” and before “In some instances they are the main driving force for protein attachment to the negatively charged plasma membrane (Cho & Stahelin, 2005)” stating “For instance, plasma membrane targeting of myristoylated

K-Ras requires a polybasic domain (Cadwallader et al., 1994; Wright & Philips, 2006). An N- terminal polybasic region localizes G protein α subunits to the plasma membrane, although this region is not required for subunit palmitoylation (Pedone & Hepler, 2007; Crouthamel et al., 2008). Other examples suggest that phosphorylation of polybasic residues located upstream of palmitoylated cysteines regulates palmitoylation of a potassium channel through an electrostatic switch (Jeffries et al., 2012).”

7) There are a few polybasic amino acids in the C-terminal part of the of SNAP-25 linker, such as R191, R198 and K20 that may also contribute to plasma membrane localization. An experiment would be optional, but at the minimum, the authors should comment on these residues.

This is a very interesting suggestion. It clarifies whether binding in general is sufficient for targeting, or whether a specific binding mode is required that positions the cysteine rich region into the proximity of the inner membrane leaflet. We have tested the mutant suggested by the referee and found that these mutations have no effect on targeting (added to Figure 1 as panel F). We state in the text “There is another small cluster of positive charges downstream of the cysteine rich region in the C-terminal part of the SNAP25 linker. Elimination of these charges by introducing the mutations R191A, R198A and K201A has no effect on membrane targeting. This indicates that the mere presence of positive charges is not sufficient. Rather, their position within the protein structure determines their effect (Figure 1).”

In the Discussion we speculate that the electrostatic contact needs to be established close to the cysteines in preparation of palmitate attachment by palmitoyl-transferases. “Elimination of positive charges distal from the cysteine cluster has no effect on targeting (Figure 1; construct SNAP*25(R191A, R198A, K201A)*. This suggests that a random electrostatic contact is not sufficient for targeting. Rather, electrostatic anchoring needs to position the cysteine(s) in a way that facilitates palmitate attachment. The electrostatic contact thus needs to be established by amino acids close to the cysteine residues.”

Reviewer #2:

This is an interesting work dealing with the determinants of membrane association and subsequent palmitoylation in SNARE proteins, with potential generality. The plasma membrane localization data and how they respond to mutation are compelling to this reviewer. The simulation methodology is not clearly specified in several key respects, and the stable association (+/1 1 peptide) of all peptide sequences tested raises questions about the simulation model and its ability to capture the desired behavior. Time until stable association is not a measure of equilibrium properties and is thus inappropriate as a metric. The equivalent metric to the experimental data is an estimate of equilibrium partition coefficient.

We agree with the referee that “*time until stable association*” is not an indicator for an equilibrium partition coefficient (which would also depend on the relative size of the system’s membrane and cytosolic components).

Since peptide secondary structure (and 3D structure also) will differ between solution and membrane-associated forms, simply constraining structure and measuring the association (or even partition) between solution and membrane-associated forms does not capture either the kinetics or equilibrium behavior of the adsorption process. Atomistic simulations of the peptides in membrane-bound and solution forms to measure structural equilibria would be required to complete this analysis.

We also agree that our MD approach is limited because secondary structure is implemented in the individual peptide models (helical conformation in aa 70 – 78 of the 35 aa long peptide). The defined MARTINI peptide topologies include a random coil section along the C-terminal half of almost all peptides. Although this allows the peptide conformation to respond to the changed environment, we will not be able to identify all potential conformational changes occurring upon membrane interaction. As our primary focus is on differences between the peptide variants in the brief moment of transition from the solvated to the membrane-attached phase, secondary structural changes taking place on longer time scales (induced by an interaction with the membrane surface) could be considered as negligible.

The authors state that they used PME electrostatics with MARTINI, but they do not state whether they also used the MARTINI polarizable water model, which is required for proper usage of PME electrostatics in the model as per the original papers. This technical point is important here, as the authors are measuring electrostatic interactions between charged peptides and a membrane (and they appear to observe artifactually stable association).

We agree with the reviewer in general. For this work we extended the standard MARTINI approach (including the standard water model) which was also used in the original publication of the plasma membrane simulation by applying PME. This allows a more realistic behaviour of (charged) peptide-membrane interactions as reported in Lee and Larson, J. Phys. Chem. B 112, 2008 and Rzepiela et al., Faraday Discuss. 144, 2009. The usage of the MARTINI polarizable water (PW) would further increase the accuracy of the simulation model. However, as the additional usage of the PW would increase the size of our already large system (>700,000 atoms) immensely (three instead of one particle per water bead would more than double the total number of particles), we opted for the compromise of PME in combination with standard MARTINI water.

In view of these issues, I would recommend that the simulations either be redone entirely or eliminated from the manuscript, as they do not provide a robust measure of the phenomena the authors are trying to predict (and indeed measure experimentally).

We have now completely eliminated the MD simulation from the manuscript.

To compensate a bit for the loss of the MD we added a wet-lab experiment for measuring (i) differences in the constructs’ binding behaviour and (ii) how binding may depend on lipids. We produced recombinant GST-tagged proteins of wt-SNAP25 (*+3*), SNAP25*-5* and SNAP25*+10* and incubated these with liposomes containing different lipids (new Figure 7).

We find that liposomes with negatively charged lipids like PS do not bind SNAP25 constructs. This changes upon addition of 4% PIP_2_ to the liposomes or 2.8% PIP_3_. The more positive charges the constructs carry, the better they bind to PIP_2/3_-containing liposomes. Further increase of PIP_3_ levels to 4% leads to even stronger binding but in a more general fashion (see new Figure 7).

In the subsection “Binding of SNAP25 constructs to liposomes” we state “Next, we tested whether membrane association of SNAP25 is directly mediated by negatively charged lipids. Therefore, we used reconstituted liposomes containing distinct lipid compositions but lacking any proteins (thus eliminating the role of potential SNAP25 binding partners such as syntaxin1) (Figure 7). […] The liposome binding assay thus identifies phosphoinositides as primary membrane targeting factors interacting with positive charges in the vicinity of the cysteine cluster. However, we cannot exclude that other sites in SNAP25 contribute to the phosphoinositide-dependent binding.”

Reviewer #3:

[…] 1) The work seems preliminary in only evaluating two SNAP25 constructs, one removing 8 Lys (SN25-5) and the other adding 4 Lys (SN25+10). The largest effects were observed with the first of these so it would be of interest to test other mutants to determine proximity to Cys quartet.

We agree that the work was preliminary in evaluating only two SNAP25 constructs (not counting the two lacking the palmitoylation sites) and we therefore added five additional constructs.

We generated two constructs in which only four lysine residues where removed, proximal (K76A, K83A, K94A and K96A) or distal of the cysteine cluster (K69A, K72A, K102A and K103A). For both constructs we observe reduced membrane targeting, but less pronounced as upon removal of eight lysines. There is no difference between the constructs. We state “We then reduced the charge from wt-SNAP25 (*+3*) to -1 by substituting four lysines with alanines (for construct details see Figure 1). […] Further reducing positive charges to a net charge of -5, by substituting all eight lysines with alanines (SNAP25*-5*; see Figure 1), abolishes targeting almost completely (Figure 1).”

Many Lys residues have acidic neighbors and it is unclear whether replacement of the Asp70, Glu73,-75 and Asp80 would be equally disruptive.

We tested the suggested construct (Asp70A, Glu73A, Glu75A and Asp80A). Despite the increase in net charge, we find diminished membrane targeting. We have thought about the observation that removal of positive charges is always accompanied by diminished targeting, while increase of positive charges only in half of the cases increases targeting. Moreover, if the positive patch is not in close proximity to the cysteine cluster it has no influence on targeting (construct SNAP25*(R191A, R198A, K201A)*; see also our reply to issue 7 of referee #1). We now discuss the following model: “[…] electrostatic anchoring needs to position the cysteine(s) in a way that facilitates palmitate attachment. […] Hence, the short targeting motif seems to be optimized to mediate electrostatic anchoring, while still allowing for access to the cysteines for attachment of palmitates.”

The work on SN25+10 is not very relevant because it does not deal with requirements in the native protein.

The SNAP25*+10* construct might not be biologically relevant. However, we find it interesting to explore the electrostatic binding mechanism both by decreasing and increasing the charges. Moreover, it is insightful that SNAP25*+10*(C-to-G) can associate with the membrane without being palmitoylated. Therefore we would prefer to include these constructs. However, if the referee insists we can remove them from the manuscript.

2) Others have suggested that hydrophobic residues are involved in targeting but this issue is not addressed or much discussed. Would replacing the same residues with hydrophobic residues preserve membrane targeting?

In a way this point is addressed in the construct SNAP25*-5* in which the hydrophilic lysines (Δt_R_ = – 23; Δt_R_ = retention time relative to glycine = 0 and phenylalanine = 100; Monera et al., J. Protein Sci. 1: 319-329 (1995)) were exchanged for the more hydrophobic alanines (Δt_R_ = 41).

We additionally generated a new construct, SNAP25*-5*_hydrophob_, in which we further increased the hydrophobicity by replacing the lysines with leucines instead of alanines. Leucines are strongly hydrophobic (Δt_R_ = 97, which is close to the maximum value of phenylalanine) and were also used in the study we believe the referee is referring to (Greaves et al., 2009), which we cited twice in the old manuscript.

In our assay, SNAP25*-5*_hydrophob_ shows no gain in targeting when compared to SNAP25*-5*. Hence, our data suggest that the hydrophobic effect cannot target the protein to the membrane independent from electrostatics. Therefore, our finding is in a way controversial to Greaves et al., 2009.

In the old manuscript we argued “…our data also show that electrostatic interactions are essential in regulating the association, in contrast to hydrophobic forces (Greaves et al., 2009), because they clearly dominate the targeting rate even in the presence of the cysteine residues.” We have extended the argument in the revised manuscript as follows: “Our data also show that hydrophobic forces (Greaves et al., 2009) are less crucial than electrostatic contacts. Although the hydrophobic amino acids in the cysteine cluster remain unchanged, removal of charges abolishes targeting, even though lysines are exchanged for more hydrophobic residues (alanines in the constructs SNAP25*-1_distal_*, SNAP25*-1_proximal_*and SNAP25*-5*, and leucines in SNAP25*-5_hydrophob_*).”

3) The conclusions of the article would be much stronger if an assessment of palmitoylation were conducted. The authors instead infer palmitoylation based on Cys to Gly.

To address this issue we developed an experiment based on click-chemistry. Palmitate carrying an alkyne group in its acyl chain was added to the cells after transfection. The expressed SNAP25 constructs are then palmitoylated with the modified palmitate. After cell lysis, a fluorescent dye (Cy5) is covalently attached to the acyl chain by click-chemistry. The incorporated dye is visualized after western blotting. Relating the dye to the amount of protein (quantified by an anti-GFP staining), we find that SNAP25*-5* is less and SNAP25*+10* is equally palmitoylated compared to wt-SNAP25 (see new Figure 5). As a control, we included SNAP25 _(C-toG)_ which cannot be pamitoylated and, as expected, is not labelled with Cy5.

In the manuscript we state “[…] we examined whether the extent of plasma membrane targeting correlates with the degree of palmitoylation for wt-SNAP25 *(+3)*, SNAP25*-5* and SNAP25*+10*, using SNAP25_(C-to-G)_ as a negative control. […] Moreover, SNAP25(C-to-G) cannot associate with the membrane. Finally, SNAP25*+10* shows a trend towards increased targeting. However, the data suggest that SNAP25*+10* not only associates with the plasma membrane through palmitoylation but also by pure protein electrostatics.”

4) Data are shown as SEM but there is no statistical analysis for Figure 1, Figure 2, Figure 3, Figure 4, Figure 5, Figure 6, Figure 8 so it is unclear what differences are significant.

We added statistical information to all figures.

5) The results shown in Figure 8 are quite puzzling. The images shown do not seem to be reflective of the intensity differences plotted in Figure 8. The left panels of 8A show a remarkable suppression of PLCPH by SN25 (but not by C to G) whereas histograms indicate 2X change. It is not clear which of these intensity differences is statistically significant. There is no indication that similar amounts of protein were being expressed in these studies in comparisons done in a cell line that has endogenous SN25. Lastly, it is not at all clear what the interpretation of these studies would be.

We repeated this experiment in a cell line lacking endogenous SNAP25, and additionally used a mutant PH-PLCδ, which cannot bind PIP_2_, as a control. We also corrected the data for the protein expression levels, and performed statistics. Please also see our reply to point #5 raised by referee #1.

6) The images of the membrane sheets (Figure 4) are confusingly shown at different scales and cannot be compared to bar graphs. These should be shown as direct comparisons to agree with the bar graphs.

This has been corrected. Please see our reply to point #3 raised by referee #1.

7) To eliminate expression level as an important variable in Figure 1, the authors provide Figure 1—figure supplement 1 to indicate lack of correlation. However, the ordinate scale greatly exceeds the range over which the results of Figure 1 were taken so the lack of correlation over this broad scale did not seem to be tailored to the experimental work shown.

We apologize for not having stated clearly in the figure legend that this supplementary figure uses the same data points that were averaged for Figure 1. Though on average the ratios range from 1.3 to 4.5 the variability between cells is high. The y-axes were extended to a ratio of ~ 30 to include two particularly high data points, and all y-axes were then similarly scaled to allow for better comparability. However, when the data points in this supplementary figure are averaged day-wise, the same values as in Figure 1 are obtained. Since we used the same imaging conditions throughout all experiments, we decided to pool the imaged cells of all experimental days of each construct into one plot. When analysing the experimental days individually instead, we do not find a correlation between intensity and membrane / cytosol ratio either. To avoid misunderstandings, we now clearly state in the figure legend of Figure 1—figure supplement 2 (formerly Figure 1—figure supplement 1) that the same data points as in the main figure are used. In addition, we included breaks in the y-axis, to visualize relationship between expression level and low membrane/cytosol ratios in greater detail.

8) Two features of Figure 5 need to be addressed. Firstly, the red and green channels exhibit very close similarity in the lower panels. Most previous studies on Syx/SN25 do not show this degree of alignment so the possibility of channel overlap should be addressed.

The crosstalk between the two channels (Channel1: BF position: 500 nm, BF range: 30 nm; Channel 2: BF position: 555nm, BF range: 60 nm) was checked using single transfections (please see Figure 9), yielding no relevant crosstalk at our imaging conditions.

Author response image 2.Crosstalk controls.(**a**) PC12 membrane sheet from SNAP25-GFP and syntaxin-RFP transfected cells. In the lower image a membrane expressing only syntaxin-RFP is shown, with no bleed-through into the green channel (see red box in the lower panel). (**b**) Single transfection with SNAP25-GFP. A membrane sheet with high amounts of SNAP25-GFP is imaged in both the GFP- and the RFP-channels. No crosstalk from the green into the red channel is noticed. The images in both (**a**) and (**b**) are provided at two different look up tables (LUT) (upper and lower row).**DOI:**
http://dx.doi.org/10.7554/eLife.19394.016

As pointed out by the referee, in most published work syntaxin and SNAP25 domains do not overlap. At first glance this is in contrast to the overlap seen in our images. However, in old Figure 5 (now Figure 6) we use very large pixels to avoid bleaching during the FRAP recordings. These pixels are too large for resolving individual clusters (please see Figure 10). The overlap is most prominent at the extremes of the membranes (because here two membrane layers overlap), with few spotty structures (in the lower panel of Figure 6) likely reflecting membrane invaginations, ruffles, or organelles (for better resolved structures see Figure 10). Of course, we always place the region for bleaching in an area that is devoid of such spots (see also the bleached square in Figure 6). We now state in the figure legend: “Please note that in this experiment large pixels were used to keep bleaching low. Therefore, the spatial resolution is lower than in the other experiments and does not allow for the resolving the SNAP25 micropatterning.”

Author response image 3.Upper panels, PC12 membrane sheet with SNAP25-GFP imaged via confocal laser scanning microscopy in a FRAP experiment (pixel size is 414 nm).Lower panels, same preparation imaged by fluorescence microscopy at a pixel size of 65 nm. The smaller pixels allow resolving the spotty SNAP25 domain pattern (lower right), which is not resolved with large pixels (upper right). In the lower left some bright spots are visible that could be organelles or other membranous structures.**DOI:**
http://dx.doi.org/10.7554/eLife.19394.017

Secondly, overexpression of Syx appeared to slow the mobility of all fluorescent objects in the lower row compared to upper row. Is the result of extreme overexpression?

Please see our reply to point #4 of referee #1 asking whether slow-down would be based on a true interaction between syntaxin and SNAP25 or might be a general phenomenon.

[Editors’ note: the author responses to the re-review follow.]

The manuscript has been improved but there are two remaining issues that need to be addressed before acceptance, as outlined below:

1) To reinforce the notion that SN25 membrane translocation requires PIP2, the authors construct the study of Figure 8 of expressing the PH domain of PLCdelta1 and finding less SN25 on membrane sheets. However, in this study the PH domain was expressed for 48 hrs, there was less SN25 expressed (compared to PHmut), and the cells are drastically altered in their morphology. The latter suggests drastic cytoskeletal changes or apoptosis that occur in depriving a cell long term of PIP2. Either of these would undoubtedly affect SN25 trafficking through an endosomal pool and its turnover in the cells. There does not seem to be a straightforward interpretation for the results of Figure 8. In the absence of more concerted characterization, this study should be removed from the manuscript.

We have removed Figure 8 from the manuscript.

*2) In Figure 7, in vitro protein-liposome interaction studies show that WT SN25 binding to liposomes is enhanced by inclusion of PIP2. The SN25-5 mutant exhibits some decrease in binding. This could be interpreted as PIP2 playing some role in recruiting SN25 to the membrane although the loss of in vitro binding by SN25-5 is much smaller than that claimed for cellular membrane interactions. This difference between the in vitro binding studies and the cellular studies should be discussed.*

In Figure 1 we have determined the difference in the periphery / cytosol ratio (which is different from the membrane associated fraction) and in Figure 3 and Figure 4 we show that the diminishment in the membrane-associated fraction ranges from ≈50% (Figure 3) to ≈75% (Figure 4). Depending on the concentration of highly charged phosphoinositides, in the liposome binding studies the effect is not observed (at 4% PIP3, which we already discussed) or ranges between ≈25% and ≈60%). In general, the outcomes of the different assays are difficult to compare in a quantitative manner, as the plasma membrane contains additional factors that may affect SNAP25 binding, and the number of binding sites or concentration of binding partners may differ. Moreover, in the cellular system palmitoylation plays a role as well.

We state: “These conditions yielded a stronger difference in binding in the range between the cell fractionation (Figure 3) and the membrane sheet assay (Figure 4). Although the magnitudes of the effects are difficult to compare due to the different assay systems (the complex composition of the plasma membrane versus simple lipid mix in liposomes, concentrations of binding partners and the presence of SNAP25-palmitoylation in the cellular experiments may modulate the outcome), the observations point to primary interactions occurring in the cysteine-rich region.”

Also, the results with SN25+10 do not seem to be relevant because this mutant has acquired additional MARCKS protein-like properties of PIP2 binding that may not be relevant to the WT SN25 protein.

We thank the reviewer for this comment and agree that the results of SNAP25*+10* may not be relevant for the behaviour of wt-SNAP25 and stated“The increased membrane binding observed after introducing additional positive charges (SNAP25 *+10*) may reflect a MARCKS protein-like PIP2-binding behaviour not physiologically relevant for the targeting of SNAP25/SNAP23.”.